# Towards Certified Defense for Unrestricted Adversarial Attacks

## Abstract

Certified defenses against adversarial examples are very important in safety-critical applications of machine learning. However, existing certified defense strategies only safeguard against perturbation-based adversarial attacks, where the attacker is only allowed to modify normal data points by adding small perturbations. In this paper, we provide certified defenses under the more general threat model of unrestricted adversarial attacks. We allow the attacker to generate arbitrary inputs to fool the classifier, and assume the attacker knows everything except the classifiers' parameters and the training dataset used to learn it. Lack of knowledge about the classifiers parameters prevents an attacker from generating adversarial examples successfully. Our defense draws inspiration from differential privacy, and is based on intentionally adding noise to the classifier's outputs to limit the attacker's knowledge about the parameters. We prove concrete bounds on the minimum number of queries required for any attacker to generate a successful adversarial attack. For a simple linear classifiers we prove that the bound is asymptotically optimal up to a constant by exhibiting an attack algorithm that achieves this lower bound. We empirically show the success of our defense strategy against strong black box attack algorithms.

## 1 Introduction

Classifiers trained using machine learning can often achieve high accuracy on test data coming from the same distribution as the training one. Unfortunately, for a class of inputs called adversarial examples (Szegedy et al., 2014), many classifiers can be fooled to give wrong predictions with high confidence. This poses a significant security risk for real world deployment of machine learning models. To mitigate this threat, many methods have been proposed to defend against adversarial examples (see, *e.g.*, Papernot et al. (2016); Song et al. (2018a); Ma et al. (2018); Buckman et al. (2018)). However, empirically grounded methods are often found to be ineffective using newer attack methods that they had originally not considered (Carlini & Wagner, 2017; Athalye et al., 2018). To end this arms race, defenses with certified theoretical guarantees against a threat model have recently gained more traction (see, *e.g.*, Raghunathan et al. (2018); Wong & Kolter (2017); Dvijotham et al. (2018); Cohen et al. (2019)).

Existing certified defense methods are all restricted to perturbation-based attacks (Song et al., 2018a), a threat model where the attacker is only allowed to generate an adversarial example by perturbing existing data points using small adversarial noise. In addition, a certified defense for one type of perturbation (e.g., based on $l_2$ norm) can still be vulnerable to a different type of perturbation (e.g. spatial transform (Xiao et al., 2018)). Following Song et al. (2018b) and Brown et al. (2018), we consider the much more general threat model of unrestricted adversarial attacks, where the small perturbation restriction is removed: *any input is considered a valid adversarial example as long as it induces the classifier to predict a different label than an oracle classifier*. As shown in Song et al. (2018b), existing certified defenses are not robust to unrestricted adversarial attacks.

In this paper, we provide a defense strategy with (asymptotically) certified guarantees against unrestricted adversarial attacks. The attacker can use any strategy to choose any input, and the attack is successful as long as the input is confidently classified into the wrong class by the target classifier. Note that in the white-box setup, where the attacker has access to all parameters of the classifier, defending against unrestricted adversarial examples is essentially impossible unless the classifier is

perfect, because the attacker can simply enumerate all possible inputs to find adversarial ones (we do not assume any computational restriction of the attacker). As a result, we focus our discussion on black-box unrestricted attacks, where the attacker does not know the exact parameters of the model nor the exact training data (otherise, the attacker could recover the parameters by simulating the learning algorithm), but can query the model for a limited number of times before making unrestricted adversarial examples. For example, for online services, it is easy to limit the number of queries allowed per user.

Consider a classifier learned using empirical risk minimization. Because the training data are randomly drawn from the data-generating distribution, the parameters of the classifier will be random even if the training algorithm is deterministic. Since we assume a black-box setting where the attacker does not know the learned parameters or the training dataset, the attacker has to obtain extra information by querying the model, even though the exact training algorithm and even the data-generating distribution might be known to the attacker. As a result, we can adopt methods similar to those used in differential privacy (Dwork et al., 2014) to reduce the amount of information about the parameters leaked from each query issued by the attacker.

We give concrete lower bounds on the minimum number of queries an attacker needs in order to successfully generate adversarial examples. For most classifiers the guarantees are asymptotic, meaning that the guarantees are valid if the number of training examples is sufficiently large. For very simple classifiers we prove finite sample guarantees. Our bound is also asymptotically minimax optimal with respect to important parameters of the problem — the dimension of the input and the size of the training set. We prove this by exhibiting a class of classifiers and a concrete attacking algorithm that successfully generates adversarial examples while querying no more than the minimum necessary according to the lower bounds (up to a constant).

Experimentally, we show that the defense strategy is effective against strong black-box unrestricted attacks, even when we use complex classifiers for which no theoretical guarantee with finite sample size is available. Compared to models without any defense, the number of queries needed to identify vulnerabilities often increases by an order of magnitude using our defense strategy, with negligible effect on the accuracy on "clean" inputs. Providing finite sample guarantees for complex models such as deep neural networks is an open problem for future work.

## 2 PROBLEM DEFINITION

### 2.1 CLASSIFICATION

Given a training dataset $\{(\mathbf{x}_i, y_i) \in \mathcal{X} \times \mathcal{Y}\}_{i=1}^n \overset{\text{i.i.d.}}{\sim} p^*(\mathbf{x})p^*(y \mid \mathbf{x})$, we consider the task of learning a classifier to predict the label $y$ for an input $\mathbf{x}$. To simplify our discussion, we assume that $\mathcal{X} = [-1, 1]^d$, and $\mathcal{Y} = \{-1, 1\}$. Let $\{g(\mathbf{w}, \cdot) : \mathcal{X} \to \mathbb{R}\}$ be a class of score functions with parameter $\mathbf{w}$, whose signs determine the class labels, i.e., $y = \text{sign}(g(\mathbf{w}, \mathbf{x}))$, where we define $\text{sign}(x) = 1$ when $x \geq 0$, and $\text{sign}(x) = -1$ otherwise. The model family is assumed to be well-specified, i.e., there exists a ground-truth parameter $\mathbf{w}^*$ such that the true labeling function is given by $y = \text{sign}(g(\mathbf{w}^*, \mathbf{x}))$.

We can use any learning algorithm to obtain $\mathbf{w}$ based on a training dataset. Since the training data are randomly generated from the underlying distribution $p^*(\mathbf{x})p^*(y \mid \mathbf{x})$, $\mathbf{w}$ is also a random vector and we denote its distribution as $p_{\text{learn}}(\mathbf{w})$. With a reasonable learning algorithm, $p_{\text{learn}}(\mathbf{w})$ should be increasingly concentrated around $\mathbf{w}^*$ while the size of the training dataset $n$ grows. When deploying the classifier to real world applications, we might want to use a (possibly stochastic) surrogate score function $f(\mathbf{w}, \cdot) : \mathcal{X} \to \mathbb{R}$ which depends on the learned parameter $\mathbf{w}$ of our model. Usually, it is just $g(\mathbf{w}, \cdot)$, but to mitigate adversarial attacks we might choose $f(\mathbf{w}, \cdot)$ to be some special classifier with a defense strategy. To avoid low confidence incorrect predictions, we give no response when the confidence is low. Specifically, we allow a threshold $\alpha \in \mathbb{R}^+$ such that given an input query $\mathbf{x}$, the predicted label is

$$y = \begin{cases} 1, & \text{if } f(\mathbf{w}, \mathbf{x}) > \alpha \\ -1, & \text{if } f(\mathbf{w}, \mathbf{x}) < -\alpha \\ \text{No response}, & \text{otherwise.} \end{cases} \tag{1}$$

---

PROTOCOL 1. THE ATTACK-DEFENSE PROTOCOL
**Attacker knowledge**: $\mathbf{w}^*$, $p_{\text{learn}}(\mathbf{w})$, $T$, $g(\cdot, \cdot)$, $f(\cdot, \cdot)$.
**Interacting procedure**:

1. The *attacker* chooses the distributions $q_t, t = 1, \cdots, T$ and $Q_{\text{attack}}$ defined as below.

2. The *defender* samples $\mathbf{w} \sim p_{\text{learn}}(\mathbf{w})$.

3. For $t \leftarrow 1, \cdots, T$

   (a) The *attacker* observed $\mathbf{o}_{1:t-1} = (\mathbf{x}_1, z_1), \cdots, (\mathbf{x}_{t-1}, z_{t-1})$ and sample $\mathbf{x}_t \sim q_t(\mathbf{x}_t | \mathbf{o}_{1:t-1})$ as the next query.

   (b) The *defender* sends $z_t = f(\mathbf{w}, \mathbf{x}_t)$ to the attacker.

4. The *attacker* samples $q \sim Q_{\text{attack}}(q \mid \mathbf{o}_{1:T})$, where $q$ is a probability distribution on $\mathcal{X}$. The attacker outputs $q(\mathbf{x})$.

---

Figure 1: The attack-defense protocol

## 2.2 PERFORMANCE EVALUATION

We evaluate the expected performance of the classifier $f(\mathbf{w}, \mathbf{x})$ on a test distribution $q(\mathbf{x})p^*(y \mid \mathbf{x})$. The traditional setting in statistical machine learning assumes that $q(\mathbf{x}) = p^*(\mathbf{x})$. In adversarial settings, however, $q(\mathbf{x})$ is the distribution of adversarial examples and is very different from $p^*(\mathbf{x})$. Since we consider unrestricted attacks, we assume no relationship between $q(\mathbf{x})$ and $p^*(\mathbf{x})$.

The classifier $f(\mathbf{w}, \mathbf{x})$ can make two kinds of errors for a given input $\mathbf{x}$. First, $f(\mathbf{w}, \mathbf{x})$ might give no response. We call this the *no-response error*, which is denoted as

$$\text{NR}(\mathbf{x}; \alpha) \triangleq \mathbb{I}(|f(\mathbf{w}, \mathbf{x})| \leq \alpha),$$

where $\mathbb{I}[\mathcal{P}]$ is the indicator function whose value is 1 when the property $\mathcal{P}$ holds; other it produces 0. Secondly, $f(\mathbf{w}, \mathbf{x})$ can produce a wrong prediction with high confidence. We call this the *margin error*, which is defined as

$$\text{ME}(\mathbf{x}; \alpha) = \mathbb{I}(g(\mathbf{w}^*, \mathbf{x}) < 0 \cap f(\mathbf{w}, \mathbf{x}) > \alpha) + \mathbb{I}(g(\mathbf{w}^*, \mathbf{x}) > 0 \cap f(\mathbf{w}, \mathbf{x}) < -\alpha).$$

An ideal classifier $f(\mathbf{w}, \mathbf{x})$ should minimize the expected no-response error $\mathbb{E}_{q(\mathbf{x})}[\text{NR}(\mathbf{x}; \alpha)]$ and margin error $\mathbb{E}_{q(\mathbf{x})}[\text{ME}(\mathbf{x}; \alpha)]$ simultaneously. When defending against an adversary, the margin errors are more important because a confident but wrong prediction is arguably more harmful than giving no prediction at all. Consequently, we only focus on bounding the margin errors achievable by attackers in the sequel. We will only consider no response error for clean data $\mathbf{x} \sim p^*(\mathbf{x})$.

## 2.3 THE ATTACK-DEFENSE PROTOCOL

In Figure 1, we provide a specific protocol of attack and defense to capture our assumptions of the threat model. Since we consider the threat model of unrestricted adversarial attacks, we assume the attacker to be fully general: The attacker is allowed to use any adaptive set of intermediate distributions $\{q_t(\mathbf{x} \mid \mathbf{o}_{1:t-1})\}_{t=1}^T$ to issue queries based on all previous observations. For the final attack distribution $q$, the attacker is allowed to sample it from any distribution of distributions $Q_{\text{attack}}(q \mid \mathbf{o}_{1:T})$. We call a specific choice of $\{q_t(\mathbf{x} \mid \mathbf{o}_{1:t-1})\}_{t=1}^T$ and $Q_{\text{attack}}$ an *attack strategy*, and a specific choice of $p_{\text{learn}}(\mathbf{w})$ and $f(\cdot, \cdot)$ a *defense strategy*. We additionally *allow the attacker to know the exact defense strategy*. Note that when the attack and defense strategies are fixed beforehand, the attack-defense protocol defines a joint probability over $\mathbf{w}, \mathbf{o}_{1:T}$ and $q$.

The protocol can be modified in the defender's favor if defender only sends $y_t$ according to Eq.(1) instead of $z_t$. However, using $z_t$ allows normal users (sampling from $p^*(\mathbf{x})$) to get a confidence score, and any certified defense results are more general (they are still true if attacker only has access to $y_t$).

Intuitively, the attack strategy is successful under the attack-defense protocol, if with high probability the final attack distribution $q$ incurs large expected margin errors for $f(\mathbf{w}, \mathbf{x})$. To formalize this intuition, we define a winning strategy of the attacker as follows.

**Definition 1.** *For a given defense strategy, an attack strategy in the attack-defense protocol is an $(\alpha, \gamma, \delta)$-winning strategy if with probability at least $\gamma$, the attacker outputs a $q$ such that $\Pr_{q(\mathbf{x})}[\text{ME}(\mathbf{x}; \alpha)] \geq \delta$.*

In the following part, we will propose a concrete defense strategy and prove that no winning strategies can exist under some conditions.

## 3 TOWARDS A CERTIFIED DEFENSE

In order for a certified defense strategy to exist we need some reasonable conditions on the problem. All the proofs are available in the appendix.

### 3.1 DEFENSIBILITY

For an omniscient attacker who knows $f(\mathbf{w}, \cdot)$ and $g(\mathbf{w}^*, \cdot)$ (this is beyond the capabilities granted by our attack-defense protocol), there will not be any effective defense strategies unless $f(\mathbf{w}, \mathbf{x})$ never produces wrong predictions. This is because the attacker can simply enumerate an input $\mathbf{x}_{\text{bad}}$ so that $|f(\mathbf{w}, \mathbf{x}_{\text{bad}}) - g(\mathbf{w}^*, \mathbf{x}_{\text{bad}})|$ is maximal, and set $q(\mathbf{x}) = \delta_{\mathbf{x}_{\text{bad}}}$ (here $\delta_{\mathbf{x}}$ denotes a point mass distribution on $\mathbf{x}$). This results in the maximal average margin error for $f(\mathbf{w}, \mathbf{x})$.

Under our attack-defense protocol, the attacker does not have access to $\mathbf{w}$ and hence do not know $f(\mathbf{w}, \cdot)$. The randomness of $p_{\text{learn}}(\mathbf{w})$ makes it more difficult for the attacker to infer $\mathbf{w}$ through queries. To guarantee the existance of a certified defense for all attacks, we need to make sure the randomness of $p_{\text{learn}}(\mathbf{w})$ is sufficient such that no fixed distribution $q(\mathbf{x})$ can incur large average margin error for $f(\mathbf{w}, \cdot)$ with high probability for a random draw of $\mathbf{w} \in p_{\text{learn}}(\mathbf{w})$. We formalize this intuition as the following condition.

**Condition 1.** *Given a classifier $g(\cdot, \cdot)$, margin $\alpha$, and true parameter $\mathbf{w}^*$, a distribution $p_{\text{learn}}(\mathbf{w})$ is $t$-defensible if $\forall \mathbf{x} \in \mathcal{X}$, we have $\mathbb{E}_{p_{\text{learn}}(\mathbf{w})}[|g(\mathbf{w}, \mathbf{x}) - g(\mathbf{w}^*, \mathbf{x})| \geq \alpha] \leq e^{-t}$.*

In other words, the above condition says that if $p_{\text{learn}}(\mathbf{w})$ is $t$-defensible, there will not exist any $\mathbf{x} \in \mathcal{X}$ such that $g(\mathbf{w}, \mathbf{x})$ significantly deviates from the ground-truth score $g(\mathbf{w}^*, \mathbf{x})$ with large probability over $\mathbf{w} \sim p_{\text{learn}}(\mathbf{w})$.

### 3.2 QUERY PRIVACY

As discussed before, the attacker cannot find an attack distribution $q$ a-priori to fool $f(\mathbf{w}, \cdot)$ when $p_{\text{learn}}(\mathbf{w})$ is $t$-defensible. To strive for successful attacks, the attacker must then obtain more information about the specific $\mathbf{w}$ used by the defender through queries. Therefore, a successful defense strategy should not leak too much information through each query from the attacker. We formalize this intuition as follows.

**Condition 2.** *A defense strategy in the attack-defense protocol is $s$-query private if for any $\mathbf{x} \in \mathcal{X}$ we have $I(z; \mathbf{w} \mid \mathbf{x}) \leq s$.*

### 3.3 CERTIFYING THE DEFENSE STRATEGY

To satisfy this condition and prevent leakage of information on $\mathbf{w}$, we take inspiration from differential privacy and perturb the scores with Gaussian noise. By adding the right amount of noise, we hope to hit the sweet spot where the accuracy on clean input $\mathbf{x} \sim p^*(\mathbf{x})$ is mostly preserved, while at the same time the attacker obtains least information from queries. We use the following stochastic surrogate function $f(\cdot, \cdot)$ as the key of our defense strategy.

$$\boxed{f(\mathbf{w}, \mathbf{x}) = g(\mathbf{w}, \mathbf{x}) + \mathcal{N}(0, \tau^2)} \tag{2}$$

As the first main result of our paper, the following theorem asserts that under the above two conditions no attacker has a winning strategy using a limited number of queries.

**Theorem 1.** *Suppose $p_{\text{learn}}(\mathbf{w})$ is $t$-defensible, and the defense strategy is $s$-private. $\forall \alpha > 0$, $0 < \gamma, \delta < 1$, and for $\tau^2 \leq \frac{\alpha^2}{8 \log 2/\delta}$, there is no $(2\alpha, \gamma, 2\delta)$ winning strategy if $T \leq \frac{1}{s}(\gamma t + \gamma \log \delta - \log 2)$.*

Note that in the definition above $s$ implicitly depends on $\tau^2$. Adding more noise (larger $\tau^2$) reduces the amount of leaked information on $\mathbf{w}$.

In practice, it can be hard to verify the defensibility of $p_{\mathrm{learn}}(\mathbf{w})$ or the query-privacy of the defense strategy for complicated models. In the following, we show examples of simple models where these two conditions are either strictly satisfied or asymptotically satisfied in the limit of infinite data. In the next section, we will first show that for logistic regression and kernel logistic regression models, the conditions hold asymptotically when the models are well-specified. After that, we show that the conditions can hold exactly for a simple naïve Bayes model. We leave the analysis of defensibility and query-privacy for more complicated models as an open problem for future research.

### 3.4 ASYMPTOTIC ANALYSIS OF (KERNEL) LOGISTIC REGRESSION

We start by considering a linear model $g(\mathbf{w}, \mathbf{x}) = (\mathbf{w}, \mathbf{x})$, where $(\cdot, \cdot)$ denotes the inner product in Euclidean space. One common approach to learning this model is logistic regression. When the model is well-specified, *i.e.*, there exists $\mathbf{w}^*$ such that $p^*(y \mid \mathbf{x}) = \frac{1}{1+\exp(-y(\mathbf{w}^*, \mathbf{x}))}$, we can prove that $p_{\mathrm{learn}}(\mathbf{w})$ converges in distribution to a Gaussian distribution with mean $\mathbf{w}^*$ as $n \to \infty$. Specifically, we have the following lemma.

**Lemma 1.** *Assume that* $\|\mathbf{w}^*\| \leq \lambda$. *Given i.i.d. samples* $\{(\mathbf{x}_i, y_i)\}_{i=1}^n \sim p^*(\mathbf{x})p^*(y \mid \mathbf{x})$, *the estimator* $\mathbf{w}_n$ *obtained by solving the following objective function*

$$\max_{\|\mathbf{w}\| \leq 2\lambda} \frac{1}{n} \sum_{i=1}^n \log(1 + e^{-y_i(\mathbf{w}, \mathbf{x}_i)})$$

*is consistent. Moreover,*

$$\sqrt{n}(\mathbf{w}_n - \mathbf{w}^*) \xrightarrow{d} \mathcal{N}(0, \Sigma),$$

*where* $\Sigma = \left(\mathbb{E}_{\mathbf{x}}[p(y = 1 \mid \mathbf{x})(1 - p(y = 1 \mid \mathbf{x}))\mathbf{x}\mathbf{x}^{\mathsf{T}}]\right)^{-1}$.

The asymptotic Gaussianity of $p_{\mathrm{learn}}(\mathbf{w})$ greatly facilitates the investigation in its defensibility. However, the assumption that a linear model $g(\mathbf{w}, \mathbf{x})$ being well-specified is rather restrictive. We therefore additionally consider a generalization called kernel logistic regression, where $g(\mathbf{w}, \mathbf{x}) = \langle \mathbf{w}, \phi(\mathbf{x}) \rangle$. Here $\phi(\mathbf{x})$ is the natural basis vector of the Reproducing Kernel Hilbert Space $\mathcal{H}$ corresponding to a kernel $k(\cdot, \cdot)$, and we use $\langle \cdot, \cdot \rangle$ to denote the inner product in $\mathcal{H}$. When the kernel is universal (Micchelli et al., 2006), *e.g.*, the Gaussian RBF kernel, functions in RKHS can approximate any bounded continuous function arbitrarily well w.r.t. the uniform norm. Therefore, it is a much milder assumption that there exists $\mathbf{w}^* \in \mathcal{H}$ such that $p^*(y \mid \mathbf{x}) = \frac{1}{1+\exp(-y\langle \mathbf{w}^*, \phi(\mathbf{x}) \rangle)}$. For kernel logistic regression, we can similarly prove that $p_{\mathrm{learn}}(\mathbf{w})$ is asymptotically Gaussian (see Lemma 6 in the appendix), with asymptotic variance

$$\Sigma_{\mathcal{H}} = \left(\mathbb{E}_{\mathbf{x}}[p(y = 1 \mid \mathbf{x})(1 - p(y = 1 \mid \mathbf{x}))\phi(\mathbf{x}) \otimes \phi(\mathbf{x})]\right)^{-1},$$

where $\otimes$ denotes the tensor product.

For both logistic regression and kernel logistic regression, under the assumptions of well-specification and in the asymptotic regime, we can prove both defensibility and query privacy. Formally,

**Proposition 1.** *For logistic regression, the asymptotic distribution of* $p_{\mathrm{learn}}(\mathbf{w})$ *is* $\frac{n\alpha^2}{2\|\Sigma\|_2 d}$*-defensible and our defense strategy is* $\frac{\|\Sigma\|_2 d}{2n\tau^2}$*-query private. For kernel logistic regression, the asymptotic distribution of* $p_{\mathrm{learn}}(\mathbf{w})$ *is* $\frac{n\alpha^2}{2\|\Sigma_{\mathcal{H}}\|_{\mathrm{op}}}$*-defensible, and our defense is* $\frac{\|\Sigma_{\mathcal{H}}\|_{\mathrm{op}}}{2n\tau^2}$*-query private.*

Therefore, our defense strategy is asymptotically certified for logistic regression and kernel logistic regression models.

### 3.5 NON-ASYMPTOTIC VERIFICATION OF ASSUMPTIONS

Asymptotic conditions hold for sufficiently large training set size $n$, but to be fully verifiable we need results that hold for finite $n$. These bounds are very difficult to obtain, and we show some bounds for very simple classifiers (naive Bayes).

Before showing the results for naive Bayes classifiers, we first show defensibility of computing averages.

**Proposition 2.** *Let $p(\mathbf{x})$ be any distribution with covariance $\Sigma$ and $\mathbf{w}^* = \mathbb{E}_{p(\mathbf{x})}[\mathbf{x}]$. Let $\mathbf{x}_1, \cdots, \mathbf{x}_n \sim p(\mathbf{x})$ and $\mathbf{w} = \frac{1}{n}\sum_i \mathbf{x}_i$, then the distribution of $\mathbf{w}$ is $\frac{n\alpha^2}{2d\|\Sigma\|_2 + 2/3d\alpha}$-defensible.*

Note that the results of this theorem is almost as good as if we knew the distribution $p(\mathbf{x})$ is Gaussian $\mathcal{N}(0, \Sigma)$. In that case, we can apply Proposition 1 and conclude that it is $\frac{n\alpha^2}{2d\|\Sigma\|_2}$-defensible.

The following Corollary shows a similar bound for a naive Bayes classifier. This is a direct outcome of the above proposition.

**Corollary 1.** *Let $p(\mathbf{x}|y = 0)$ and $p(\mathbf{x}|y = 1)$ be distributions with covariance $\Sigma$, and $\mathbf{w}^* = \mathbb{E}_{p(\mathbf{x}|y=1)}[\mathbf{x}] - \mathbb{E}_{p(\mathbf{x}|y=0)}[\mathbf{x}]$. Let $\mathbf{x}_1, \cdots, \mathbf{x}_n \sim p(\mathbf{x}|y = 0)$ and $\mathbf{x}'_1, \cdots, \mathbf{x}'_n \sim p(x|y = 1)$. If we choose $w = \frac{1}{n}\sum_i \mathbf{x}_i - \frac{1}{n}\sum_i \mathbf{x}'_i$, then it is $\frac{n\alpha^2}{4d\|\Sigma\|_2 + 4/3d\alpha}$-defensible.*

Therefore, we can use the training data to upper bound $\|\Sigma\|_2$ (El Karoui et al., 2008) and obtain a non-asymptotic certificate of defensibly.

For query privacy non-asymptotic guarantees are easier to get. All we need is to upper bounding the covariance spectral radius of $p_{\text{learn}}(\mathbf{w})$.

**Proposition 3.** *If $g(\mathbf{w}, \mathbf{x}) = (\mathbf{w}, \mathbf{x})$ and $p_{\text{learn}}(\mathbf{w})$ has covariance $\Sigma$, it is $\frac{d\|\Sigma\|_2}{2\tau^2}$-query private.*

## 4 RATE ANALYSIS AND MINIMAX OPTIMALITY

To demonstrate more concrete asymptotic rates, we will analyze a simple setup. We study a classifier with $g(\mathbf{w}, \mathbf{x}) = (\mathbf{w}, \mathbf{x})$ and $p_{\text{learn}}(\mathbf{w}) = \mathcal{N}(\mathbf{w}^*, \sigma^2 I)$ where $\sigma^2 = 1/n$. We also assume that $\|\mathbf{w}^*\|_\infty < b$ for some $b > 0$ (we need $b$ to be fixed as we analyze the asymptotic rate when $d \to \infty$). We give an example of a classifier where this distribution could arise in the appendix. Based on this setup, we have the following specialization of Theorem 1.

**Corollary 2.** *There is no $(2\alpha, \gamma, 2\delta)$ winning strategy if $T < \frac{\alpha^4 n^2 \gamma}{2d^2 \log 2/\delta} + o(n)$ if we choose $\tau^2 = \frac{\alpha^2}{2 \log 2/\delta}$.*

This bound is desirable if $\alpha \gg 1$ (there is a large margin) and $n \gg d$ (more data than input dimensions).

### 4.1 AN ATTACKER STRATEGY

Corollary 2 is the minimum number of queries any attacker strategy needs to win. Now we show a concrete attacker strategy that (almost) achieves this lower bound.

The concrete attacker strategy is shown below. Intuitively, the attacker queries each of the $d$ dimensions of the input $\mathbf{x} \in \mathbb{R}^d$ to find out the sign of $\mathbf{w} - \mathbf{w}^*$. The attacker finds a $\mathbf{x}$ such that $(\mathbf{x}, \mathbf{w} - \mathbf{w}^*)$ is large but $(\mathbf{x}, \mathbf{w}^*) < 0$.

---

PROTOCOL 2. THE SIGN ATTACK PROTOCOL

1. *Defender* samples $\mathbf{w} \sim p_{\text{learn}}(\mathbf{w})$.

2. For round $= t = 1, \cdots, T$

   (a) *Attacker* chooses $\mathbf{x}_t = \epsilon_{t \bmod d}$, where $\epsilon_i$ is the $i$-th basis vector for $\mathbb{R}^d$.
   (b) *Defender* sends $z_t = (\mathbf{x}_t, \mathbf{w}) + \mathcal{N}(0, \tau^2)$ to attacker.

3. *Attacker* computes $\bar{\mathbf{w}}_i = \text{avg}\{z_t | \mathbf{x}_t = \epsilon_i\}$. Without loss of generality (by re-indexing the dimensions) $|\mathbf{w}_1^*| > |\mathbf{w}_2^*| > \cdots > |\mathbf{w}_d^*|$. For $i = 1, \cdots, \lambda d$ choose $\mathbf{x}_i = -\text{sign}(\mathbf{w}_i^*)$, and for $i = \lambda d, \cdots, d$ choose $\mathbf{x}_i = \text{sign}(\bar{\mathbf{w}}_i - \mathbf{w}_i^*)$. Attacker chooses the smallest $\lambda$ such that $(\mathbf{x}, \mathbf{w}^*) < 0$. She outputs a delta distribution on this $\mathbf{x}$ as his attack distribution.

---

**Theorem 2.** *For any $0 < \gamma < 1$, $\alpha > 0$, if $d$ is sufficiently large, the attacker in Protocol 2 is a $\left(\alpha, 1, \frac{1-\gamma}{2}\right)$ winning strategy if $T = \frac{20\alpha^4 n^2}{d^2}$ if $n \leq \frac{d^2}{160\alpha^2 \log 2/\delta}$ and we choose $\tau^2 = \frac{\alpha^2}{2\log 2/\delta}$; it is a $\left(\alpha, 1, \frac{1-\gamma}{2}\right)$ winning strategy when $T = 2\alpha\sqrt{n}$ if we choose $\tau^2 = 0$.*

The extra condition $n \leq \frac{d^2}{160\alpha^2 \log 2/\delta}$ ensures that adversarial examples actually exist; if the classifier is sufficiently accurate (i.e. $\|\mathbf{w} - \mathbf{w}^*\|_1 \leq \alpha$), then no example will be falsely classified.

## 4.2 RATE SUMMARY AND COMPARISON

Note that Corollary 2 is a lower bound on required number of queries: no attacker can have high success rate unless $T > \frac{\alpha^4 n^2 \gamma}{2d^2 \log 2/\delta}$; while Theorem 2 is an upper bound: with $T = \frac{20\alpha^4 n^2}{d^2}$ there is at least one attacker that can achieve high success rate. The two bounds have the same asymptotic with respect to $\alpha, n$ and $d$, which shows that the bound is asymptotically optimal with respect to these parameters.

To better study the asymptotic behavior for large $d$ and large $n$, we will choose $\alpha = \sqrt{d}$. For both methods $\Pr_{p^*(\mathbf{x})}[\mathrm{NR}(\mathbf{x})]$ can be arbitrarily small for sufficiently large $d$. This is proved in the Appendix. We have the following rates for the smallest $T$ such that an attacker can have a $\left(\alpha, 1, \frac{1-\gamma}{2}\right)$ winning strategy for any $\gamma > 0$ and $\alpha = \sqrt{d}$. With this choice of $\alpha$, $T$ no longer depend on $d$ for both upper and lower bounds, and we can get:

- For defended classifier with $\tau^2 = \frac{\alpha^2}{2\log 2/\delta}$ we have $T = \Theta(n^2)$ queries.
- For undefended classifier with $\tau^2 = 0$ we have $T = O(n)$.

## 5 EXPERIMENTS

### 5.1 LINEAR MODELS

We will first verify our method empirically on linear classification tasks. We will use two classification datasets.

**Synthetic Gaussian**. For this dataset we randomly sample $\mathbf{w}^*$ from a standard Gaussian, sample $p(\mathbf{x}|y=1) = w^* + \mathcal{N}(0, \tilde{\sigma}^2 I)$ and $p(\mathbf{x}|y=0) = -\mathbf{w}^* + \mathcal{N}(0, \tilde{\sigma}^2 I)$ and finally clip $\mathbf{x}$ to between $[-1, 1]$. We choose $\tilde{\sigma}$ such that $\mathbf{w}^*$ is the optimal logistic regression weight for $p(y|\mathbf{x}) = \frac{1}{1 + e^{-(\mathbf{w}^*, \mathbf{x})}}$.

**Mnist**. Because linear models cannot classify MNIST with high accuracy, we use deep network features (last layer of a pretrained AlexNet) as the input $\mathcal{X}$ instead of the pixel space. We train with the entire dataset and use the resulting vector as $\mathbf{w}^*$ (for the attacker).

**Attack Methods**. We will use two attacking method: the first one is the attack strategy of Protocol 2 (which we will denote the sign attack); the second one is Simba (Guo et al., 2019), a recently proposed black-box attacking algorithm with very good empirical performance.

**Classifiers**. We will use two classifiers, the naive Bayes classifier defined in Corollary 1 and logistic regression. For both classifiers we choose $\alpha$ such that the no-response rate on $p^*(\mathbf{x})$ (estimated using the validation dataset) is no more than 10%. We either choose $\tau^2 = 0$ (undefended) or $\tau^2 = \frac{\alpha^2}{2\log 2/\delta}$ as in Theorem 1 (defended) where $\delta$ is chosen to be 5%.

### 5.1.1 RESULTS

The results are shown in Figure 2. For defended classifier, the number of queries required for successful attack typically increases by an order of magnitude. In addition, the effect on the clean data $p^*(\mathbf{x})$ is almost negligible. Another interesting observation is that, the naive Bayes classifier actually performs much better in terms of adversarial robustness than logistic regression. We conjecture that this is because logistic regression has high parameter estimation variance $\mathrm{tr}(\mathrm{Cov}(\mathbf{w}))$ compared to naive Bayes. This indicates that adversarial robustness has very different requirements compared to classification accuracy. Even though logistic regression usually performs better in classification unless the training data size is tiny (Ng & Jordan, 2002), its adversarial robustness is poor.

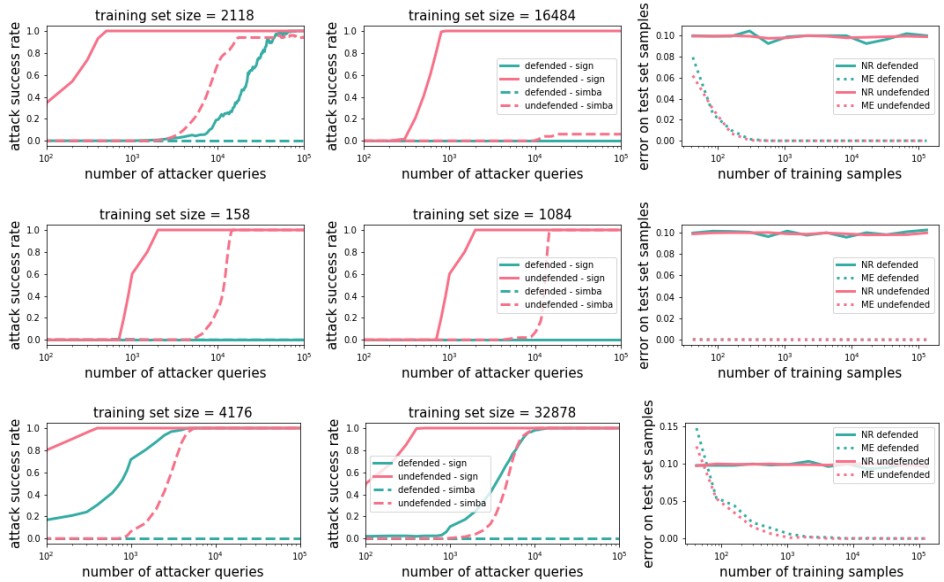

Figure 2: Success rate of attacker vs. number of queries on synthetic dataset and MNIST. Left to right: **Left and Middle**: the number of attacker queries vs. attack success rate for different training set sizes. **Right**: the no-answer (NR) and margin error (ME) on clean data $x \sim p^*(x)$. Top to bottom: **Top**: naive Bayes classifier trained on synthetic Gaussian; **Middle**: naive Bayes classifier trained on MNIST; **Bottom**: logistic regression trained on synthetic Gaussian. Logistic regression trained on MNIST fails to achieve robustness for the training set sizes we experimented (up to 10k). For both cases, defended classifiers require orders of magnitude more queries for successful adversarial attack, while the adverse effect on clean data $x \sim p^*(x)$ is negligible.

## 5.2 DEEP MODELS

Our theoretical guarantees are far from effective for deep models. It is unlikely the analysis will be effective for current network architectures, so special architectures and analysis will be necessary for certified defense under out framework. Nevertheless we perform simple experiments on deep networks to empirically verify that our defense strategy can effectively protect our model against black box attack methods. The setup is identical to (Guo et al., 2019) on ImageNet, except we add a threshold $\alpha$ to the output sigmoid probabilities: if no class has a predicted probability greater than the threshold $\alpha$, the model answers "I don't know". Here we choose $\alpha = 0.7$.

The results are shown in Figure 3. Similar to linear models, defended models require orders of magnitude more queries to attack. We hope these promising experimental performance will motivate further theoretical analysis and empirical investigation.

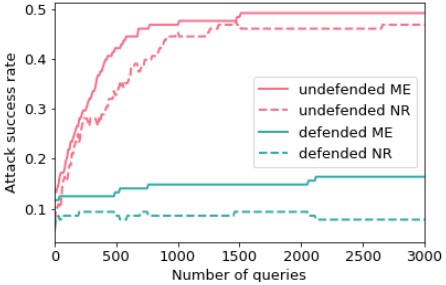

Figure 3: Success rate vs. number of queries for Simba attacker on Inception network. We adjust the threshold such that both methods have similar no response rate (NR) on the clean data. The undefended network is significantly more vulnerable to attack.

## 6 DISCUSSION

In this paper we propose a new type of adversarial defense guarantee, one based on the concept of defensibility and query privacy. We show theoretical guarantees for simple classifiers and good empirical performance for complex classifiers.

Several open questions remain to be answered. The first one to verify defensibility and query privacy for more complex classifiers. The second one is to design classifiers with better defensibility and query privacy properties. The third one is large scale empirical investigation of the defense strategy or similar defense strategies, i.e. studying the performance of adding noise to different layers of a deep neural network against a larger suite of black box attack algorithms. We hope our work can serve as a first step in this exciting direction of research.

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

# A   APPENDIX

## A.1   PROOFS RELATED TO THE LOWER BOUND OF $T$

*Proof of Theorem 1.* If a strategy is $(2\alpha, \gamma, 2\delta)$ winning then with probability $\gamma$

$$\Pr_{q(\mathbf{x})}[|g(\mathbf{w}, \mathbf{x}) - g(\mathbf{w}^*, \mathbf{x}) + \mathcal{N}(0, \tau^2)| \geq 2\alpha] \geq 2\delta \tag{3}$$

Because $|g(\mathbf{w}, \mathbf{x}) - g(\mathbf{w}^*, \mathbf{x}) + \mathcal{N}(0, \tau^2)| \leq |g(\mathbf{w}, \mathbf{x}) - g(\mathbf{w}^*, \mathbf{x})| + |\mathcal{N}(0, \tau^2)|$. If Eq.(3) is true one of the following must be true

$$\Pr[|\mathcal{N}(0, \tau^2)| \geq \alpha] \geq \delta \tag{4}$$

$$\Pr_{q(\mathbf{x})}[|g(\mathbf{w}, \mathbf{x}) - g(\mathbf{w}^*, \mathbf{x})| \geq \alpha] \geq \delta \tag{5}$$

For Eq.(4) we can bound by

$$\Pr[|\mathcal{N}(0, \tau^2)| \geq \alpha] \leq 2e^{-\frac{\alpha^2}{2\tau^2}}$$

Therefore, Eq.(4) is false if the RHS is less than $\delta$, which is when $\tau^2 \leq \frac{\alpha^2}{2\log 2/\delta}$.

now we show the condition for Eq.(5) to be true. We first define the following notation.

- $\mathcal{X}(\mathbf{w}) = \{\mathbf{x} : |g(\mathbf{w}, \mathbf{x}) - g(\mathbf{w}^*, \mathbf{x})| \geq \alpha\}$

- $M(\mathbf{x}) = \{\mathbf{w} : |g(\mathbf{w}, \mathbf{x}) - g(\mathbf{w}^*, \mathbf{x})| \geq \alpha\}$

- Let $q$ be any distribution on $\mathcal{X}$, let $q(\mathcal{X}(\mathbf{w}))$ denote the probability mass $q$ assigns to $\mathcal{X}(\mathbf{w})$.

The following lemma is needed.

**Lemma 2.** *In out attack-defense protocol, if with probability at least $\gamma$, we have $\Pr[|g(\mathbf{w}, \mathbf{x}) - g(\mathbf{w}^*, \mathbf{x})| \geq \alpha] \geq \delta$, then $I(\mathbf{w}; \mathbf{o}_{1:T}) \geq \gamma(t + \log \delta) - \log 2$.*

Therefore, if $I(\mathbf{w}; \mathbf{o}_{1:T}) < \gamma(t + \log \delta) - \log 2$, Eq. (5) is false with at least $\gamma$ probability. If we additionally have the condition that the attack-defense protocol is $s$-private, by a sequence of data processing inequalities, and by the fact that $\mathbf{x}_t$ only depends on $\mathbf{w}$ through $\mathbf{o}_{<t}$, and $y_t$ only depends on $\mathbf{x}_t$ and $\mathbf{w}$, we have

$$I(\mathbf{w}; \mathbf{o}_{1:T}) = I(\mathbf{w}; \mathbf{o}_{<T}) + I(\mathbf{w}; \mathbf{x}_T | \mathbf{o}_{<T}) + I(\mathbf{w}; y_T | \mathbf{x}_T, \mathbf{o}_{<T}) \leq I(\mathbf{w}; \mathbf{o}_{<T}) + I(\mathbf{w}; y_T | \mathbf{x}_T)$$

$$\leq \cdots \leq \sum_{t=1}^{T} I(\mathbf{w}; y_t | \mathbf{x}_t) \leq Ts$$

Therefore Eq. (5) is false with at least $\gamma$ probability if

$$I(y_t; \mathbf{w} | \mathbf{x}_T) \leq sT < \gamma(t + \log \delta) - \log 2$$

which is equivalent to $T < \frac{1}{s}(\gamma t + \gamma \log \delta - \log 2)$. What remains is to prove Lemma 2, which we do below.

*Proof of Lemma 2.* By the condition of $t$ defensibility we have

$$\mathbb{E}_{p_{\text{learn}}(\mathbf{w})}[q(\mathcal{X}(\mathbf{w}))] = \mathbb{E}_{p_{\text{learn}}(\mathbf{w})}\mathbb{E}_{q(\mathbf{x})}[\mathbb{I}(\mathbf{x} \in \mathcal{X}(\mathbf{w}))] = \mathbb{E}_{q(\mathbf{x})}[\mathbb{I}(\mathbf{w} \in M(\mathbf{x}))]$$

$$= \mathbb{E}_{q(\mathbf{x})}\mathbb{E}_{p_{\text{learn}}(\mathbf{w})}[\mathbb{I}(\mathbf{w} \in M(\mathbf{x}))] \leq \mathbb{E}_{q(\mathbf{x})}[e^{-t}] = e^{-t}$$

We abbreviate $\mathbb{E}_{p_{\text{learn}}(\mathbf{w})}$ by $\mathbb{E}_{\mathbf{w}}$. Define $s(q; \mathbf{w}) = \max(q(\mathcal{X}(\mathbf{w})), e^{-t})$, then

$$\mathbb{E}_{\mathbf{w}}[s(q; \mathbf{w})] \leq \mathbb{E}_{\mathbf{w}}[q(\mathcal{X}(\mathbf{w}))] + \mathbb{E}_{\mathbf{w}}[e^{-t}] \leq 2e^{-t} = e^{-t+\log 2}$$

In addition we have $\mathbb{E}_{\mathbf{w}}[\mathbb{E}_{Q(q)}[s(q; \mathbf{w}))]] = \mathbb{E}_{Q(q)}[\mathbb{E}_{\mathbf{w}}[s(q; \mathbf{w})]] \leq e^{-t+\log 2}$. Finally by Donsker-Varadhan inequality,

$$\mathbb{E}_{\mathbf{w}}\mathbb{E}_{Q(q|\mathbf{w})}[\log s(q; \mathbf{w})] \leq \mathbb{E}_{\mathbf{w}}\text{KL}(Q(q \mid \mathbf{w}) \| Q(q)) + \mathbb{E}_{\mathbf{w}}\log \mathbb{E}_{Q(q)}[e^{\log s(q;\mathbf{w})}]$$

$$\leq I(q; \mathbf{w}) + \log \mathbb{E}_{\mathbf{w}}\mathbb{E}_{Q(q)}[s(q; \mathbf{w})]$$

$$\leq I(q; \mathbf{w}) - t + \log 2$$

Suppose $\delta > e^{-t}$, $\log q(\mathcal{X}(\mathbf{w})) \geq \log \delta$ if and only if $\log s(q; \mathbf{w}) \geq \log \delta$. In addition, because for all $q$ and $\mathbf{w}$, $\log s(q; \mathbf{w}) \geq -t$, so $\log s(q; \mathbf{w}) + t$ is a non-negative random variable, and we can apply Markov inequality to get (where the probability and expectation are taken with respect to $\mathbf{w} \sim p_{\text{learn}}(\mathbf{w})$ and $q \sim Q(q \mid \mathbf{w})$)

$$\gamma = \Pr[q(\mathcal{X}(\mathbf{w})) \geq \delta] = \Pr\left[\log q(\mathcal{X}(\mathbf{w})) \geq \log \delta\right] = \Pr[\log s(q; \mathbf{w}) \geq \log \delta]$$

$$= \Pr[\log s(q; \mathbf{w}) + t \geq \log \delta + t] \leq \frac{\mathbb{E}[\log s(q; \mathbf{w}) + t]}{\log \delta + t} \leq \frac{I(q; \mathbf{w}) + \log 2}{\log \delta + t}$$

Rearranging we get

$$I(q; \mathbf{w}) \geq \gamma(t + \log \delta) - \log 2$$

Finally because $q$ is a function of $o_{1:T}$, we have by data processing

$$I(\mathbf{w}; \mathbf{o}_{1:T}) \geq I(\mathbf{w}; q) \geq \gamma(t + \log \delta) - \log 2$$

$\square$

$\square$

*Proof of Corollary 2.* First by Lemma 4 we know that $p_{\text{learn}}(\mathbf{w})$ is $\frac{\alpha^2}{2\sigma^2 d}$-defensible and $\frac{\sigma^2 d}{2\tau^2}$-query private. We apply Theorem 1 and choose $\tau^2 = \frac{\alpha^2}{2\log 2/\delta}$ to conclude that there is no $(2\alpha, \gamma, 2\delta)$ winning strategy if

$$T \leq \frac{2\tau^2}{\sigma^2 d}\left(\gamma \frac{\alpha^2}{2\sigma^2 d} + \gamma \log \delta - \log 2\right) = \frac{\alpha^2 n}{d \log 2/\delta}\left(\gamma \frac{\alpha^2 n}{2d} + \gamma \log \delta - \log 2\right)$$

$\square$

## A.2 PROOFS RELATED TO THE SIGN ATTACK

*Proof of Theorem 2.* Under the distribution in Protocol 2, denote

$$\hat{\mathbf{x}} = (\hat{\mathbf{x}}_1, \cdots, \hat{\mathbf{x}}_d) = (\text{sign}(\bar{\mathbf{w}}_1 - \mathbf{w}_1^*), \cdots, \text{sign}(\bar{\mathbf{w}}_d - \mathbf{w}_d^*))$$

Also denote the probability of attacker producing $\hat{\mathbf{x}}$ given forecaster choose $\mathbf{w}$ as $q(\hat{\mathbf{x}} \mid \mathbf{w})$. By observing the protocol we know that this distribution is factorized, i.e. $q(\hat{\mathbf{x}} \mid \mathbf{w}) = q(\hat{\mathbf{x}}_1 \mid \mathbf{w}) \cdots q(\hat{\mathbf{x}}_d \mid \mathbf{w})$. Also denote $\mathbf{b}^* = \text{sign}(\mathbf{w} - \mathbf{w}^*)$ as the true sign of the error.

**Step 1:** We show that if $q(\hat{\mathbf{x}}_i = b_i^* \mid \mathbf{w}) \geq 1/2 + \beta$, $(\hat{\mathbf{x}}, \mathbf{r}^*)$ is likely large

**Lemma 3.** *Under the distribution defined in Protocol 2, suppose for all $i$ we have $q(\hat{\mathbf{x}}_i = b_i^* \mid \mathbf{w}) \geq 1/2 + \beta$, then*

$$\Pr\left[(\hat{\mathbf{x}}, \mathbf{r}^*) \geq \frac{\sqrt{2}}{\sqrt{\pi}}\beta d\sigma\right] \geq 1 - e^{-2\beta d/\pi}$$

*Proof of Lemma 3.* The proof idea is to first lower bound the expectation, then lower bound the deviation from the expectation.

$$\mu := \mathbb{E}[(\hat{\mathbf{x}}, \mathbf{r}^*)] = \sum_i \mathbb{E}[\hat{\mathbf{x}}_i \mathbf{r}_i^*] = \sum_i \mathbb{E}[\hat{\mathbf{x}}_i \mathbf{r}_i^* \mid \mathbf{x}_i = b_i^*]\Pr[\hat{\mathbf{x}}_i = b_i^*] + \mathbb{E}[\hat{\mathbf{x}}_i \mathbf{r}_i^* \mid \hat{\mathbf{x}}_i \neq b_i^*]\Pr[\hat{\mathbf{x}}_i \neq b_i^*]$$

$$\geq \sum_i \mathbb{E}[|\mathbf{r}_i^*|](1/2 + \beta) - \mathbb{E}[|\mathbf{r}_i^*|](1/2 - \beta) = 2\beta\mathbb{E}[\|r^*\|_1] = \frac{2\sqrt{2}}{\sqrt{\pi}}\beta d\sigma$$

Then because $(\hat{\mathbf{x}}, \mathbf{r}^*) = \sum_i \hat{\mathbf{x}}_i \mathbf{r}_i^*$, where $\{\hat{\mathbf{x}}_i \mathbf{r}_i^*, i = 1, \cdots, d\}$ are independent mixtures of Gaussian. In addition $\hat{\mathbf{x}}_i \mathbf{r}_i^*$ has a sub-Gaussian tail

$$\Pr[\hat{\mathbf{x}}_i \mathbf{r}_i^* \geq c] \leq \Pr[|\mathbf{r}_i^*| \geq c] \leq e^{-\frac{c^2}{4\sigma^2}}$$

so it is $\sigma^2/2$-sub Gaussian. Combined we have $(\hat{\mathbf{x}}, \mathbf{r}^*)$ is $\sigma^2 d/2$-sub Gaussian. Therefore

$$\Pr[(\hat{\mathbf{x}}, \mathbf{r}^*) \leq \mu - cd] \leq e^{-\frac{c^2 d^2}{d\sigma^2}} = e^{-\frac{c^2 d}{\sigma^2}}$$

If we plug in $c = \frac{\sqrt{2}}{\sqrt{\pi}}\beta\sigma$ we have

$$\Pr\left[(\hat{\mathbf{x}}, \mathbf{r}^*) \leq \frac{\sqrt{2}}{\sqrt{\pi}}\beta d\sigma\right] \leq e^{-\frac{2\beta d}{\pi}}$$

$\square$

**Step 2: Lower bound on** $q(\hat{\mathbf{x}}_i = \mathbf{b}_i^* \mid \mathbf{w})$. The first step is to show that $\hat{\mathbf{x}}_i = \mathbf{b}_i^*$ with non-negligible probability. Because $\bar{\mathbf{w}}_i - \mathbf{w}_i^* \sim \mathcal{N}(\mathbf{r}_i^*, \tau^2/T)$ and if $\tau^2/T \geq 2\sigma^2$

$$\Pr[\hat{\mathbf{x}}_i = \mathbf{b}_i^*] = \Pr[\mathcal{N}(0, \tau^2/T) \leq |\mathcal{N}(0, \sigma^2)|] \geq \frac{1}{2} + \frac{\sigma\sqrt{T}}{10\tau} \qquad (6)$$

The proof of the above statement relies on the following Lemma

**Lemma 4.** *Let* $x \sim \mathcal{N}(0, a^2)$, $y \sim \mathcal{N}(0, b^2)$ *be independent random variables, then if* $a \geq 2b$, *then* $\frac{1}{2} + \frac{b}{10a} \leq \Pr[x \leq |y|] \leq \frac{1}{2} + \frac{b}{\pi a}$.

*Proof of Lemma 4.*

$$\Pr[x \leq |y|] = \mathbb{E}_y\left[\Pr[x \leq |y| | y]\right] = \mathbb{E}_y\left[\Phi\left(\frac{|y|}{a}\right)\right]$$

Because $\Phi$ is concave in $[0, +\infty)$ we have

$$\mathbb{E}_y\left[\Phi\left(\frac{|y|}{a}\right)\right] \leq \mathbb{E}_y\left[\frac{|y|}{a}\Phi'(0) + \frac{1}{2}\right] = \frac{1}{a}\frac{1}{\sqrt{2\pi}}\mathbb{E}_y[|y|] + \frac{1}{2} = \frac{1}{\sqrt{2\pi}a}\frac{b\sqrt{2}}{\sqrt{\pi}} + \frac{1}{2} = \frac{b}{\pi a} + \frac{1}{2}$$

For a lower bound we have for any $u \geq 0$,

$$\mathbb{E}_y\left[\Phi\left(\frac{|y|}{a}\right)\right] \geq \frac{1}{2} + \mathbb{E}_y\left[\mathbb{I}(|y| \leq ub)\left(|y|\frac{\Phi(ub/a) - 1/2}{ub}\right)\right] + \mathbb{E}_y\left[\mathbb{I}(|y| > ub)\left(\Phi\left(\frac{ub}{a}\right) - \frac{1}{2}\right)\right]$$

$$\geq \frac{1}{2} + \frac{\Phi(ub/a) - 1/2}{ub}\mathbb{E}_y\left[\mathbb{I}(|y| \leq ub)|y|\right]$$

According to Wikipedia for truncated Gaussians, we know that $\mathbb{E}_y[y|0 < y < ub] = b\frac{\phi(0) - \phi(u)}{\Phi(u) - \Phi(0)}$, so

$$\mathbb{E}_y\left[\mathbb{I}(|y| \leq ub)|y|\right] = \mathbb{E}_y\left[|y| \mid |y| < ub\right]\Pr[|y| < ub] = b\frac{\phi(0) - \phi(u)}{\Phi(u) - \Phi(0)}2(\Phi(u) - \Phi(0))$$

$$= 2b(\phi(0) - \phi(u)) = \frac{2b}{\sqrt{2\pi}}(1 - e^{-u^2/2}) \equiv g(u)b$$

In addition by the concavity of $\Phi$ on $[0, +\infty)$ we have

$$\Phi\left(\frac{ub}{a}\right) \geq \frac{1}{2} + \Phi'\left(\frac{ub}{a}\right)\frac{ub}{a} \geq \frac{1}{2} + \phi\left(\frac{ub}{a}\right)\frac{ub}{a}$$

Combining the above results we have

$$\mathbb{E}_y\left[\Phi\left(\frac{|y|}{a}\right)\right] \geq \frac{1}{2} + \frac{\Phi(ub/a) - 1/2}{ub}g(u)b \geq \frac{1}{2} + g(u)\phi\left(\frac{ub}{a}\right)\frac{b}{a}$$

Now we can plug in numerical results. Let $u = 1$, and if $b/a \leq 1/2$ we get the results in the Lemma. $\square$

**Step 3: Bound the agreement between x and $\hat{\mathbf{x}}$.** note that $\hat{\mathbf{x}}$ can have high value of $(\hat{\mathbf{x}}, \mathbf{r}^*)$, but this is not useful unless in the final $\mathbf{x}$ outputted, a large proportion of the dimensions is equal to $\hat{\mathbf{x}}$. In other words, we need to show that $\lambda$ cannot be too large.

**Lemma 5.** *In Protocol 2,* $\forall s > 0$, $\lambda < \frac{s\sqrt{bd} + b}{d}$ *with probability* $1 - e^{-s^2/2}$, *where b is an upper bound on* $\|\mathbf{w}^*\|_\infty$.

To simplify our analysis, we will assume that $d$ is large enough such that $\lambda \leq 1/2$ with probability $1 - \gamma/2$.

*Proof of Lemma 5.* We know that $\|\mathbf{w}^*\|_\infty < b$ and $\|w\|_1 = d$, then by Hölder inequality have $\|\mathbf{w}^*\|_2^2 = (\mathbf{w}^*, \mathbf{w}^*) \leq \|\mathbf{w}^*\|_1 \|\mathbf{w}^*\|_\infty \leq bd$.

For any choice of $s > 0$ because the distribution of $\mathbf{x}_{\lambda d:d}$ do not depend on $\mathbf{w}^*_{\lambda d:d}$, and by symmetry arguments $\mathbf{x}_i, i = \lambda d, \cdots, d$ are i.i.d. Bernoulli distributions with zero mean, so they are $1/2$ sub-Gaussian. We know that $(\mathbf{x}_{\lambda d:d}, \mathbf{w}^*_{\lambda d:d})$ is $\sum_{i=\lambda d}^{d} \frac{(\mathbf{w}_i)^2}{2} \leq \frac{\|\mathbf{w}^*\|_2^2}{2} \leq \frac{bd}{2}$ sub-Gaussian. We can conclude for any $s > 0$

$$\Pr[(\mathbf{x}_{\lambda d:d}, \mathbf{w}^*_{\lambda d:d}) \geq s\sqrt{bd}] \leq e^{-s^2/2} \tag{7}$$

Because the assumption (without loss of generality) that $|\mathbf{w}_1^*| \geq |\mathbf{w}_2^*| \geq \cdots \geq |\mathbf{w}_d^*|$ we know that $\|\mathbf{w}^*_{1:\lambda d}\|_1 \geq \lambda \|\mathbf{w}^*\|_1 \geq \lambda d$, so $(\mathbf{x}_{1:\lambda d}, \mathbf{w}^*_{1:\lambda d}) = -\|\mathbf{w}^*_{1:\lambda d}\|_1 < -\lambda d$, so with probability at least $1 - e^{-s^2/2}$

$$(\mathbf{x}, \mathbf{w}) = (\mathbf{x}_{1:\lambda d}, \mathbf{w}^*_{1:\lambda d}) + (\mathbf{x}_{\lambda d:d}, \mathbf{w}^*_{\lambda d:d}) \leq -\lambda d + s\sqrt{bd}$$

note that the algorithm will choose the smallest $\lambda$ such that $(\mathbf{x}, \mathbf{w}) < 0$. In particular, it will not choose a $\lambda$ such that $(\mathbf{x}, \mathbf{w}) < -b$, so we have with probability at least $1 - e^{-s^2/2}$, $-\lambda d + s\sqrt{bd} \geq -b$, or $\lambda \leq \frac{s\sqrt{bd}+b}{d}$. $\qquad\square$

**Step 4: Combining the Results.** Finally we are in a position to bound $\mathrm{ME}(\mathbf{x})$ and prove Theorem 2.

We know that $\forall i = 1, \cdots, \lambda d$, the distribution of $\mathbf{r}_i^*$ is independent of the choice of $\mathbf{x}_i$, $(\mathbf{x}_{1:\lambda d}, \mathbf{r}^*_{1:\lambda d})$ is $\sum_{i=1}^{\lambda d} \mathbf{x}_i^2 \sigma^2 \leq \lambda d \sigma^2$ sub-Gaussian, so for any $s > 0$ we have

$$\Pr[(\mathbf{x}_{1:\lambda d}, \mathbf{r}^*_{1:\lambda d}) \geq s\sqrt{\lambda d}\sigma] \leq e^{-s^2/2} \tag{8}$$

By Lemma 3 we have

$$\Pr\left[(\mathbf{x}_{\lambda d:d}, \mathbf{r}^*_{\lambda d:d}) \geq \frac{\sqrt{2}}{\sqrt{\pi}}\beta(1-\lambda)d\sigma\right] \geq 1 - e^{2\beta(1-\lambda)d/\pi}$$

Combined we have

$$\Pr\left[(\mathbf{x}, \mathbf{r}^*) \geq \frac{\sqrt{2}}{\sqrt{\pi}}\beta(1-\lambda)d\sigma - s\sqrt{\lambda d}\sigma\right] \geq 1 - e^{2\beta(1-\lambda)d/\pi} - e^{-s^2/2}$$

And if we plug in the above condition that $\lambda < 1/2$ (which is true with probability $1 - \gamma/2$) we have

$$\Pr\left[(\mathbf{x}, \mathbf{r}^*) \geq \frac{\sqrt{2}}{2\sqrt{\pi}}\beta d\sigma - s\sqrt{2d}\sigma\right] \geq 1 - e^{\beta d/\pi} - e^{-s^2/2}$$

By previous analysis we know $\beta \geq \frac{\sigma\sqrt{T}}{10\tau}$, so we can choose a sufficiently large $s$ and given the choice of $s$ we can choose a sufficiently large $d$ such that

$$\Pr\left[(\mathbf{x}, \mathbf{r}^*) \geq \frac{1}{2\sqrt{\pi}}\frac{\sigma\sqrt{T}}{10\tau}d\sigma\right] \geq 1 - \gamma/2$$

So if $\frac{1}{2\sqrt{\pi}}\frac{\sigma\sqrt{T}}{10\tau}d\sigma \geq \alpha$ we have

$$\Pr[(\mathbf{x}, \mathbf{r}^*) \geq \alpha] \geq 1 - \gamma$$

note that $\mathrm{ME}(\mathbf{x}) = 1$ whenever $(\mathbf{x}, \mathbf{r}^*) + \mathcal{N}(0, \tau^2) \geq \alpha$. Because of the symmetry of the Gaussian distribution, this happens with $1/2$ probability if $(\mathbf{x}, \mathbf{r}^*) \geq \alpha$. Therefore we have

$$\Pr[\mathrm{ME}(\mathbf{x}) = 1] \geq \frac{1}{2}(1 - \gamma)$$

We can rewrite the condition as

$$T \geq \frac{400\pi\alpha^2 n^2 \tau^2}{d^2} = \frac{400\pi\alpha^4 n^2}{64d^2}$$

In particular we can choose $T = \frac{20\alpha^4 n^2}{d^2}$. This is true if the condition if Lemma 4 is true. That is $\tau/\sqrt{T} \geq 2\sigma$, or $T \leq \tau^2/4\sigma^2 = \frac{\alpha^2 n}{8\log 2/\delta}$. Such a $T$ could exist if

$$\frac{\alpha^2 n}{8\log 2/\delta} \geq \frac{20\alpha^4 n^2}{d^2} \qquad n \leq \frac{d^2}{160\alpha^2 \log 2/\delta}$$

Conversely if $\tau = 0$, we have $\mathbb{E}[\|\mathbf{r}_{1:T}\|_1] = \frac{\sqrt{2}T\sigma}{\sqrt{\pi}}$. In addition by Lemma 5 for large $d$, $\lambda \to 0$, so when $T < d$ we have (when $d$ is sufficiently large)

$$\Pr\left[(\mathbf{x}, \mathbf{r}^*) \geq \frac{T\sigma}{\sqrt{\pi}}\right] = \Pr\left[\|\mathbf{r}_{1:T}\|_1 \geq \frac{T\sigma}{\sqrt{\pi}}\right] \to_{d\to\infty} 1$$

We can similarly show that $\Pr[\text{ME}(\mathbf{x}) = 1] \geq \frac{1}{2}(1 - \gamma)$ if $T\sigma/\sqrt{\pi} \geq \alpha$, which becomes the condition $T \geq \sqrt{\pi}\alpha\sqrt{n}$. We choose $T = 2\alpha\sqrt{n}$ to satisfy it.

$\square$

## A.3 Proofs related to the conditions

**Proposition 4** (Representer Theorem with Hard Constraints). *The optimal solution*

$$w_n \triangleq \arg\min_{\|w_n\| \leq 2\lambda} \frac{1}{n}\sum_{i=1}^{n}\log(1 + e^{-y_i\langle w, \phi(x_i)\rangle})$$

*can be represented as*

$$w_n = \sum_{i=1}^{n}\alpha_i\phi(x_i),$$

*where $\alpha_i$ is a function of $\{(x_i, y_i)\}_{i=1}^{n}$.*

*Proof of Proposition 4.* We first make a critical observation that $l(\mathbf{x}, y; \mathbf{w}) = \log(1 + e^{-y\langle \mathbf{w}, \phi(\mathbf{x})\rangle})$ is convex w.r.t. $\mathbf{w}$. We can then use the convex duality theory to obtain the following equivalent optimization problem.

$$\max_{\alpha}\min_{w}\frac{1}{n}\sum_{i=1}^{n}\log(1 + e^{-y_i\langle \mathbf{w}, \phi(\mathbf{x}_i)\rangle}) + \mu(\|\mathbf{w}\| - 2\lambda), \quad \mu \geq 0.$$

Due to the traditional representer theorem, we know that $\mathbf{w}^* = \sum_{i=1}^{n}\alpha_i(\mu^*)\phi(\mathbf{x}_i)$, where $(\mathbf{w}^*, \mu^*)$ is a saddle point of the max-min problem. This completes our proof. $\square$

*Proof of Proposition 1.* For logistic regression, first we observe that because $\|\mathbf{x}\|_\infty \leq 1$, it must be that $\|\mathbf{x}\|_2 \leq \sqrt{d}$. Let $\Sigma = Q^T\Lambda Q$ where $Q$ is orthonomal and $\Lambda$ is diagonal. Because $Q$ is orthonormal, $\|Q\mathbf{x}\|_2 \leq \sqrt{d}$, then

$$\mathbf{x}^T\Sigma\mathbf{x} \leq \sup_{y:\|y\|_2^2=d} y^T\Lambda y = \sup_{y:\|y\|_2^2=d}\sum_i \Lambda_{ii}y_i^2 \leq \left(\sup_i \Lambda_{ii}\right)\left(\sum_i y_i^2\right) = \|\Lambda\|_2 d = \|\Sigma\|_2 d$$

now we can bound defensibility by $(\mathbf{x}, \mathbf{w} - \mathbf{w}^*) \sim \mathcal{N}(0, \mathbf{x}^T\Sigma\mathbf{x}/n)$ and

$$\Pr\left[\mathcal{N}\left(0, \frac{\mathbf{x}^T\Sigma\mathbf{x}}{n}\right) \geq \alpha\right] \leq \Pr\left[\mathcal{N}\left(0, \frac{\|\Sigma\|_2 d}{n}\right) \geq \alpha\right] \leq e^{-\frac{n\alpha^2}{2\|\Sigma\|_2 d}}$$

We can also bound mutual information by

$$I(z; \mathbf{w} \mid \mathbf{x}) = H(z|\mathbf{x}) - H(z|\mathbf{w}, \mathbf{x}) = H(\mathcal{N}(0, \mathbf{x}^T\Sigma\mathbf{x}/n + \tau^2)) - H(\mathcal{N}(0, \tau^2))$$

$$= \frac{1}{2}\log\left(\frac{\mathbf{x}^T\Sigma\mathbf{x}/n + \tau^2}{\tau^2}\right) \leq \frac{1}{2}\log\left(1 + \frac{\|\Sigma\|_2 d}{n\tau^2}\right) \leq \frac{\|\Sigma\|_2 d}{2n\tau^2}$$

We can prove similar results for kernel logistic regression. note that $\Sigma = \Lambda^{-1}$, where

$$\Lambda \triangleq \mathbb{E}_{p(\mathbf{x})}[p(y = 1 \mid \mathbf{x})p(y = -1 \mid \mathbf{x})\phi(\mathbf{x}) \otimes \phi(\mathbf{x})] \in \mathcal{H} \otimes \mathcal{H}.$$

We will first show that $\Lambda$ is a bounded operator. This is because for all $f \in \mathcal{H}$, we have

$$
\begin{aligned}
\|\Lambda f\| &= \|\mathbb{E}[p(y = 1 \mid \mathbf{x})p(y = -1 \mid \mathbf{x})\phi(\mathbf{x})\langle f, \phi(\mathbf{x})\rangle]\| \\
&\leq \mathbb{E}[\|p(y = 1 \mid \mathbf{x})p(y = -1 \mid \mathbf{x})\phi(\mathbf{x})\langle f, \phi(\mathbf{x})\rangle\|] \\
&\leq \mathbb{E}[p(y = 1 \mid \mathbf{x})p(y = -1 \mid \mathbf{x})\|\phi(\mathbf{x})\|^2\|f\|] \\
&\leq \mathbb{E}[p(y = 1 \mid \mathbf{x})p(y = -1 \mid \mathbf{x})]\|f\| \\
&\leq \frac{\|f\|}{4}.
\end{aligned}
$$

Equivalently, $\|\Lambda\|_{\text{op}} \leq \frac{1}{4}$. Due to the Bounded Inverse Theorem, $\Sigma = \Lambda^{-1}$ is also a bounded operator and therefore $\|\Sigma\|_{\text{op}} < \infty$. Therefore, for any $x \in \mathcal{X}$,

$$\langle \phi(\mathbf{x}), \Sigma\phi(\mathbf{x})\rangle \leq \|\phi(\mathbf{x})\|\|\Sigma\phi(\mathbf{x})\| \leq \|\Sigma\|_{\text{op}}.$$

As a result,

$$\Pr[\mathcal{N}(0, \langle\phi(\mathbf{x}), \Sigma\phi(\mathbf{x})\rangle) \geq \alpha n] \leq \Pr\left[\mathcal{N}\left(0, \frac{\|\Sigma\|_{\text{op}}}{n}\right) \geq \alpha\right] \leq e^{-\frac{n\alpha^2}{2\|\Sigma\|_{\text{op}}}}.$$

The mutual information can be bounded by

$$
\begin{aligned}
I(z; \mathbf{w} \mid \mathbf{x}) &= H(z|\mathbf{x}) - H(z|\mathbf{w}, \mathbf{x}) = H(\mathcal{N}(0, \langle\phi(\mathbf{x}), \Sigma\phi(\mathbf{x})\rangle/n + \tau^2)) - H(\mathcal{N}(0, \tau^2)) \\
&= \frac{1}{2}\log\frac{\langle\phi(\mathbf{x}), \Sigma\phi(\mathbf{x})\rangle/n + \tau^2}{\tau^2} \leq \frac{1}{2}\log\left(1 + \frac{\|\Sigma\|_{\text{op}}}{n\tau^2}\right) \leq \frac{\|\Sigma\|_{\text{op}}}{2n\tau^2}
\end{aligned}
$$

$\square$

*Proof of Proposition 2.* We know that for any $\mathbf{x}_0 \in \mathcal{X}$

$$
\begin{aligned}
\mathbb{E}[(\mathbf{x}_0, \mathbf{r})^2] &= \mathbb{E}\left[\frac{1}{n^2}\left(\sum_i(\mathbf{x}_0, \mathbf{x}_i)\right)^2\right] = \mathbb{E}\left[\frac{1}{n^2}\sum_{i,j}(\mathbf{x}_0, \mathbf{x}_i)(\mathbf{x}_0, \mathbf{x}_j)\right] \\
&= \mathbb{E}\left[\frac{1}{n^2}\sum_i(\mathbf{x}_0, \mathbf{x}_i)^2\right] + \mathbb{E}\left[\frac{1}{n^2}\sum_{i\neq j}\mathbf{x}_0^T\mathbf{x}_i\mathbf{x}_j^T\mathbf{x}_0\right] \\
&= \frac{1}{n}\mathbb{E}[(\mathbf{x}_0, \mathbf{x})^2] + \frac{1}{n^2}\sum_{i\neq j}\mathbf{x}_0^T\mathbb{E}[\mathbf{x}_i]\mathbb{E}[\mathbf{x}_j]^T\mathbf{x}_0 = \frac{1}{n}\mathbb{E}[\mathbf{x}_0^T\mathbf{x}\mathbf{x}^T\mathbf{x}_0] = \frac{1}{n}\mathbf{x}_0^T\Sigma\mathbf{x}_0 \leq \frac{d}{n}\|\Sigma\|_2
\end{aligned}
$$

Because $\mathbb{E}[(\mathbf{x}_0, \mathbf{r})] = 0$, $\text{Var}[(\mathbf{x}_0, \mathbf{r})] = \mathbb{E}[(\mathbf{x}_0, \mathbf{r})^2] \leq \frac{d}{n}\|\Sigma\|_2$, by Bernstein inequality

$$\Pr[(\mathbf{x}_0, \mathbf{r}) \geq \alpha] \leq e^{-\frac{n\alpha^2}{2d\|\Sigma\|_2 + 2/3d\alpha}}$$

$\square$

*Proof of Proposition 3.* Proof of this proposition utilizes the fact that Gaussian distributions have highest entropy among distributions with the same covariance. Let $z = (w, x) + \mathcal{N}(0, \tau^2)$, because $\text{Var}[z|x] = x^T\Sigma x + \tau^2 \leq d\rho(\Sigma) + \tau^2$, so $H(z|x) \leq \frac{1}{2}\log(d\rho(\Sigma) + \tau^2)$.

$$I(z; w|x) = H(z|x) - H(z|x, w) \leq \frac{1}{2}\log\frac{d\rho(\Sigma) + \tau^2}{\tau^2} \leq \frac{d\rho(\Sigma)}{2\tau^2}$$

$\square$

*Proof of Lemma 1.* Let us denote $l(\mathbf{x}, y; \mathbf{w}) = \log(1 + e^{-y(\mathbf{w}, \mathbf{x})})$. In order to learn $\mathbf{w}$, we minimize the negative likelihood function $P_n l(\mathbf{x}, y; \mathbf{w})$, and denote $\mathbf{w}_n \triangleq \arg\min P_n l(\mathbf{x}, y; \mathbf{w})$. We will first prove that $\mathbf{w}_n \xrightarrow{p} \mathbf{w}$. As the first step, we prove the uniform law of large numbers of $l(\mathbf{x}, y; \mathbf{w})$. notice that

$$\mathbb{E}[\sup_{\|\mathbf{w}\| \leq 2\lambda} |P_n l(\mathbf{x}, y; \mathbf{w}) - P l(\mathbf{x}, y; \mathbf{w})|]$$

$$= \mathbb{E}[\sup_{\|\mathbf{w}\| \leq 2\lambda} |\mathbb{E}[P_n l(\mathbf{x}, y; \mathbf{w}_n) - P_n l(\mathbf{x}', y'; \mathbf{w})]|]$$

$$\leq \mathbb{E}[\sup_{\|\mathbf{w}\| \leq 2\lambda} |P_n l(\mathbf{x}, y; \mathbf{w}) - l(\mathbf{x}', y'; \mathbf{w})|]$$

$$= \mathbb{E}\left[\frac{1}{n} \sup_{\|\mathbf{w}\| \leq 2\lambda} \left|\sum_{i=1}^n \epsilon_i (l(\mathbf{x}_i, y_i; \mathbf{w}) - l(\mathbf{x}'_i, y'_i; \mathbf{w}^*))\right|\right]$$

$$\leq \frac{2}{n} \mathbb{E}\left[\sup_{\|\mathbf{w}\| \leq 2\lambda} \left|\sum_{i=1}^n \epsilon_i l(\mathbf{x}_i, y_i; \mathbf{w})\right|\right]$$

$$\overset{(i)}{\leq} \frac{4}{n} \mathbb{E}\left[\sup_{\|\mathbf{w}\| \leq 2\lambda} \left|\sum_{i=1}^n \epsilon_i (\mathbf{w}, \mathbf{x}_i)\right|\right] = \frac{4}{n} \mathbb{E}\left[\sup_{\|\mathbf{w}\| \leq 2\lambda} \left|(\mathbf{w}, \sum_{i=1}^n \epsilon_i \mathbf{x}_i)\right|\right]$$

$$\overset{(ii)}{\leq} \frac{8\lambda}{n} \mathbb{E}\left[\left\|\sum_{i=1}^n \epsilon_i \mathbf{x}_i\right\|\right] \overset{(iii)}{\leq} \frac{8\lambda}{n} \sqrt{\mathbb{E}\left[\left\|\sum_{i=1}^n \epsilon_i \mathbf{x}_i\right\|^2\right]}$$

$$= \frac{8\lambda}{\sqrt{n}},$$

where $(i)$ is due to the Ledoux-Talagrand contraction theorem, $(ii)$ is due to Cauchy-Schwarz, and $(iii)$ is due to Jensen's inequality. Using Markov's inequality, we can conclude that with probability $1 - \delta$,

$$\sup_{\|\mathbf{w}\| \leq 2\lambda} |P_n l(\mathbf{x}, y; \mathbf{w}) - P l(\mathbf{x}, y; \mathbf{w})| \leq \frac{8\lambda}{\sqrt{n}\delta}.$$

Therefore, with probability $1 - \delta$, $P l(\mathbf{x}, y; \mathbf{w}_n) - P l(\mathbf{x}, y; \mathbf{w}^*)$ can be bounded as

$$0 \leq P l(\mathbf{x}, y; \mathbf{w}_n) - P l(\mathbf{x}, y; \mathbf{w}^*)$$
$$= P l(\mathbf{x}, y; \mathbf{w}_n) - P_n l(\mathbf{x}, y; \mathbf{w}_n) + P_n l(\mathbf{x}, y; \mathbf{w}_n) - P_n l(\mathbf{x}, y; \mathbf{w}^*) + P_n l(\mathbf{x}, y; \mathbf{w}^*) - P l(\mathbf{x}, y; \mathbf{w}^*)$$
$$\leq \frac{16\lambda}{\sqrt{n}\delta}.$$

Furthermore, the smoothness of $P l(\mathbf{x}, y; \mathbf{w})$ guarantees that when $\|\mathbf{w}_n - \mathbf{w}^*\|$ is small enough,

$$P l(\mathbf{x}, y; \mathbf{w}_n) - P l(\mathbf{x}, y; \mathbf{w}^*) = \nabla P l(\mathbf{x}, y; \mathbf{w}^*)^\intercal (\mathbf{w}_n - \mathbf{w}^*)$$

$$+ \frac{1}{2}(\mathbf{w}_n - \mathbf{w}^*)^\intercal \nabla^2 P l(\mathbf{x}, y; \mathbf{w}^*)(\mathbf{w}_n - \mathbf{w}^*) + o(\|\mathbf{w}_n - \mathbf{w}^*\|)$$

$$\geq \frac{1}{4}\lambda_{min}(\nabla^2 P l(\mathbf{x}, y; \mathbf{w}^*)) \|\mathbf{w}_n - \mathbf{w}^*\|^2$$

$$\overset{(i)}{\geq} \frac{1}{4}\lambda_{min}(\mathrm{Cov}[\nabla P l(\mathbf{x}, y; \mathbf{w}^*)]) \|\mathbf{w}_n - \mathbf{w}^*\|^2,$$

where $(i)$ is due to Bartlett's identity, and $\lambda_{min}$ denotes the smallest eigenvalue. Combining the fact that $P l(\mathbf{x}, y; \mathbf{w}_n) \xrightarrow{p} P l(\mathbf{x}, y; \mathbf{w}^*)$ and $P l(\mathbf{x}, y; \mathbf{w}_n) - P l(\mathbf{x}, y; \mathbf{w}^*) \geq O(\|\mathbf{w}_n - \mathbf{w}^*\|^2)$, we can conclude that with probability $1 - \delta$, there exists a large enough $n$ such that $\forall n \geq n$ :

$$\|\mathbf{w}_n - \mathbf{w}^*\|^2 \leq \frac{64\lambda}{\sqrt{n}\delta\lambda_{min}(\mathrm{Cov}[\nabla P l(\mathbf{x}, y; \mathbf{w}^*)])},$$

and as a simple corollary,

$$\mathbf{w}_n \xrightarrow{p} \mathbf{w}^*.$$

Because $w_n$ minimizes $P_n l(\mathbf{x}, y; \mathbf{w})$, we have that whenever $\|\mathbf{w}_n - \mathbf{w}^*\| < \lambda$, $\nabla P_n l(\mathbf{x}, y; \mathbf{w}_n) = 0$. This is equivalent to saying that $\sqrt{n} \nabla P_n l(\mathbf{x}, y; \mathbf{w}_n) \xrightarrow{p} 0$. Using Taylor expansion, we obtain $\nabla P_n l(\mathbf{x}, y; \mathbf{w}^*) + \nabla^2 P_n l(\mathbf{x}, y; \mathbf{w}^*)(\mathbf{w}_n - \mathbf{w}^*) + o_p(1)(\mathbf{w}_n - \mathbf{w}^*) = \nabla P_n l(\mathbf{x}, y; \mathbf{w}_n) = o_p(\frac{1}{\sqrt{n}})$, which is equivalent to

$$\sqrt{n}(P \nabla^2 l(\mathbf{x}, y; \mathbf{w}^*) + (P_n - P) \nabla^2 l(\mathbf{x}, y; \mathbf{w}^*) + o_p(1))(\mathbf{w}_n - \mathbf{w}^*) = -\sqrt{n} P_n \nabla l(\mathbf{x}, y; \mathbf{w}^*) + o_p(1).$$

Combining the central limit theorem and Slutsky's theorem, we have

$$\sqrt{n}(\mathbf{w}_n - \mathbf{w}^*) \xrightarrow{d} \mathcal{N}(0; \mathbb{E}[\nabla^2 l(\mathbf{x}, y; \mathbf{w}^*)]^{-1} \operatorname{Cov}[\nabla l(\mathbf{x}, y; \mathbf{w}^*)] \mathbb{E}[\nabla^2 l(\mathbf{x}, y; \mathbf{w}^*)]^{-1}).$$

Bartlett's identity asserts that $\mathbb{E}[\nabla^2 l(\mathbf{x}, y; \mathbf{w}^*)] = \operatorname{Cov}[\nabla l(\mathbf{x}, y; \mathbf{w}^*)]$. Therefore, the asymptotic variance of $\sqrt{n}(\mathbf{w}_n - \mathbf{w}^*)$ can be simplified to

$$\Sigma \triangleq \operatorname{Cov}[\nabla l(\mathbf{x}, y; \mathbf{w}^*)]^{-1} = \mathbb{E}\left[\frac{\mathbf{x}\mathbf{x}^{\mathsf{T}}}{(1 + e^{y(\mathbf{w}^*, \mathbf{x})})^2}\right]^{-1}$$

$$= \left(\mathbb{E}_{\mathbf{x}}\left[\frac{e^{(\mathbf{w}^*, \mathbf{x})}}{(1 + e^{(\mathbf{w}^*, \mathbf{x})})^2} \mathbf{x}\mathbf{x}^{\mathsf{T}}\right]\right)^{-1}$$

$$= (\mathbb{E}_{\mathbf{x}}[p_{\mathbf{w}^*}(1 - p_{\mathbf{w}^*}) \mathbf{x}\mathbf{x}^{\mathsf{T}}])^{-1},$$

where $p_{\mathbf{w}^*} = \frac{1}{1 + e^{-(\mathbf{w}^*, \mathbf{x})}}$. $\qquad\qquad\square$

**Lemma 6.** *Assume that there exists* $\mathbf{w}^* \in \mathcal{H}$ *such that* $p^*(y \mid \mathbf{x}) = \frac{1}{1 + \exp(-y\langle \mathbf{w}^*, \phi(\mathbf{x}) \rangle)}$. *Assume further that* $\|\mathbf{w}^*\| \leq \lambda$, *and* $\forall \mathbf{x} : k(\mathbf{x}, \mathbf{x}) \leq 1$. *Suppose we are given a dataset* $\{(\mathbf{x}_i, y_i)\}_{i=1}^n \overset{i.i.d.}{\sim} p^*(\mathbf{x}) p^*(y \mid \mathbf{x})$. *Let* $\hat{\mathbf{w}}_n$ *be the solution to the following objective*

$$\min_{\|\mathbf{w}\| \leq 2\lambda} \frac{1}{n} \sum_{i=1}^n \log(1 + e^{-y_i\langle \mathbf{w}, \phi(\mathbf{x}_i) \rangle}).$$

*Then we have* $\mathbf{w}_n \xrightarrow{p} \mathbf{w}^*$, *and*

$$\sqrt{n}(\mathbf{w}_n - \mathbf{w}^*) \xrightarrow{d} \mathcal{N}(0, \Sigma_{\mathcal{H}}),$$

*where*

$$\Sigma_{\mathcal{H}} = (\mathbb{E}_{\mathbf{x}}[p(y = 1 \mid \mathbf{x})(1 - p(y = 1 \mid \mathbf{x}))\phi(\mathbf{x}) \otimes \phi(\mathbf{x})])^{-1}.$$

*Proof of Lemma 6.* Let us denote $l(\mathbf{x}, y; \mathbf{w}) = \log(1 + e^{-y\langle \mathbf{w}, \phi(\mathbf{x}) \rangle})$, $L_n(\mathbf{w}) = \frac{1}{n} \sum_{i=1}^n l(\mathbf{x}_i, y_i; \mathbf{w})$, and $L(\mathbf{w}) = \mathbb{E}[l(\mathbf{x}, y; \mathbf{w})]$. Now, we prove the uniform convergence of $L_n$ to $L$. Denote $\mathcal{B} \triangleq \{\mathbf{w} \mid \|\mathbf{w}\|_{\mathcal{H}} \leq 2\lambda\}$, and consider the expected error

$$\mathbb{E}\left[\sup_{w \in \mathcal{B}} |\hat{L}_n(\mathbf{w}) - L(\mathbf{w})|\right] = \mathbb{E}\left[\sup_{w \in \mathcal{B}} |\hat{L}_n(\mathbf{w}) - \mathbb{E}[l(\mathbf{x}, y; \mathbf{w})]|\right]$$

$$= \mathbb{E}\left[\sup_{w \in \mathcal{B}} \left|\frac{1}{n}\sum_{i=1}^n l(\mathbf{x}_i, y_i; \mathbf{w}) - \mathbb{E}\left[\frac{1}{n}\sum_{i=1}^n l(\mathbf{x}_i', y_i'; \mathbf{w})\right]\right|\right]$$

$$\leq \mathbb{E}\left[\sup_{w \in \mathcal{B}} \frac{1}{n}\left|\sum_{i=1}^n \epsilon_i[l(\mathbf{x}_i, y_i; \mathbf{w}) - l(\mathbf{x}_i', y_i'; \mathbf{w})]\right|\right]$$

$$\leq 2\mathbb{E}\left[\sup_{w \in \mathcal{B}} \frac{1}{n}\left|\sum_{i=1}^n \epsilon_i l(\mathbf{x}_i, y_i; \mathbf{w})\right|\right]$$

$$= 2\mathcal{R}[l(\mathbf{x}, y; \mathbf{w})]$$

Now let $X(\mathbf{w}) \triangleq \frac{1}{n} \sum_{i=1}^n \epsilon_i l(\mathbf{x}_i, y_i; \mathbf{w})$. note that $l(\mathbf{x}, y; \mathbf{w})$ is a 1-Lipschitz function w.r.t. $y\langle \mathbf{w}, \phi(\mathbf{x}) \rangle$. Using the Ledoux-Talagrand contraction theorem, we conclude that $\mathcal{R}(l(\mathbf{x}, y; \mathbf{w})) \leq$

$2\mathcal{R}(y\langle \mathbf{w}, \phi(\mathbf{x})\rangle)$, which can be bounded by noticing that

$$\mathcal{R}(y\langle \mathbf{w}, \phi(\mathbf{x})\rangle) = \mathbb{E}\left[\sup_{w\in\mathcal{B}} \left|\frac{1}{n}\sum_{i=1}^{n}\epsilon_i y_i\langle \mathbf{w}, \phi(\mathbf{x}_i)\rangle\right|\right]$$

$$= \mathbb{E}\left[\sup_{w\in\mathcal{B}} \left|\langle \mathbf{w}, \frac{1}{n}\sum_{i=1}^{n}\epsilon_i y_i\phi(\mathbf{x}_i)\rangle\right|\right]$$

$$\leq \mathbb{E}\left[\|\mathbf{w}\|_{\mathcal{H}} \left\|\frac{1}{n}\sum_{i=1}^{n}\epsilon_i y_i\phi(\mathbf{x}_i)\right\|_{\mathcal{H}}\right]$$

$$\leq \sqrt{\frac{\lambda^2}{n^2}\mathbb{E}\left[\sum_{i=1}^{n}\sum_{j=1}^{n}\epsilon_i\epsilon_j y_i y_j k(\mathbf{x}_i, \mathbf{x}_j)\right]}$$

$$= \frac{\sqrt{\lambda^2}}{n}\sqrt{\mathbb{E}\left[\sum_{i=1}^{n}k(\mathbf{x}_i, \mathbf{x}_i)\right]}$$

$$\leq \frac{\lambda}{\sqrt{n}},$$

where we assume that $k(\mathbf{x}, \mathbf{x}) \leq 1$, which is satisfied by many universal kernels, such as Gaussian RBF. Taken together, we have

$$\mathbb{E}\left[\sup_{w\in\mathcal{B}}|\hat{L}_n(\mathbf{w}) - L(\mathbf{w})|\right] \leq 4\sqrt{\frac{\lambda}{n}}.$$

With Markov inequality, we have $\forall \delta > 0$,

$$\Pr[\sup_{w\in\mathcal{B}}|\hat{L}_n(\mathbf{w}) - L(\mathbf{w})| > \delta] \leq \frac{4}{\delta}\sqrt{\frac{\lambda}{n}}.$$

The uniform convergence concludes that $L(\mathbf{w}_n) \xrightarrow{p} L(\mathbf{w}^*)$. Using the same arguments as in the proof of Lemma 1, we conclude that $w_n \xrightarrow{p} w^*$.

next, we prove the asymptotic normality of $w_n$. First, we note that $l(\mathbf{x}, y; \mathbf{w})$ is Fréchet differentiable w.r.t. $\mathbf{w}$. This can be verified by the chain rule of Fréchet derivatives, because $l(\mathbf{x}, y; \mathbf{w})$ is Fréchet differentiable w.r.t. $\langle \mathbf{w}, \phi(\mathbf{x})\rangle$, and $\langle \mathbf{w}, \phi(\mathbf{x})\rangle$ is Fréchet differentiable w.r.t. $\mathbf{w}$. Then, we need to focus on the differentiability of

$$\nabla_{\mathbf{w}}l(\mathbf{x}, y; \mathbf{w}) = -\frac{y\langle \phi(\mathbf{x}), \cdot\rangle}{1 + e^{y\langle \mathbf{w}, \phi(\mathbf{x})\rangle}}.$$

Let $g(\mathbf{w}) \triangleq \nabla_{\mathbf{w}}l(\mathbf{x}, y; \mathbf{w}) : \mathcal{H} \to \mathcal{H}$. We can further take the Gâteaux derivative of $g(\mathbf{w})$ to get

$$\frac{dg(\mathbf{w}_0 + t(\mathbf{w} - \mathbf{w}_0))}{dt} = \frac{e^{y\langle \mathbf{w}_0 + t(\mathbf{w}-\mathbf{w}_0), \phi(\mathbf{x})\rangle}}{(1 + e^{y\langle \mathbf{w}_0 + t(\mathbf{w}-\mathbf{w}_0), \phi(\mathbf{x})\rangle})^2}\langle \mathbf{w} - \mathbf{w}_0, \phi(\mathbf{x})\rangle\langle \phi(\mathbf{x}), \cdot\rangle$$

and even compute higher-order Gâteaux derivatives

$$\frac{d^2g(\mathbf{w}_0 + t(\mathbf{w} - \mathbf{w}_0))}{dt^2} = -\frac{e^{y\langle \mathbf{w}_0 + t(\mathbf{w}-\mathbf{w}_0), \phi(\mathbf{x})\rangle}\left(e^{y\langle \mathbf{w}_0 + t(\mathbf{w}-\mathbf{w}_0), \phi(\mathbf{x})\rangle} - 1\right)}{\left(e^{y\langle \mathbf{w}_0 + t(\mathbf{w}-\mathbf{w}_0), \phi(\mathbf{x})\rangle} + 1\right)^3}\langle \mathbf{w} - \mathbf{w}_0, \phi(\mathbf{x})\rangle^2\langle \phi(\mathbf{x}), \cdot\rangle.$$

Using fundamental theorem of calculus for Gâteaux derivatives, we obtain

$$g(\mathbf{w}) = g(\mathbf{w}_0) + \int_0^1 \frac{dg(\mathbf{w}_0 + t(\mathbf{w} - \mathbf{w}_0))}{dt}\bigg| dt$$

$$= g(\mathbf{w}_0) + \int_0^1 \frac{dg(\mathbf{w}_0 + t(\mathbf{w} - \mathbf{w}_0))}{dt}\bigg|_{t=0} dt + \int_0^1\int_0^v \frac{d^2g(\mathbf{w}_0 + t(\mathbf{w} - \mathbf{w}_0))}{dt^2}\bigg|_{t=u} dudv.$$

note that
$$\left\|\int_0^1 \int_0^v \frac{d^2 g(\mathbf{w}_0 + t(\mathbf{w} - \mathbf{w}_0))}{dt^2}\bigg|_{t=u} dudv\right\| \leq \frac{1}{20}\left\|\langle\mathbf{w} - \mathbf{w_0}, \phi(\mathbf{x})\rangle^2 \langle\phi(\mathbf{x}), \cdot\rangle\right\| = o(\|\mathbf{w} - \mathbf{w}^*\|),$$
we have
$$g(\mathbf{w}) = g(\mathbf{w}_0) + \frac{e^{y\langle\mathbf{w_0}, \phi(\mathbf{x})\rangle}}{(1 + e^{y\langle\mathbf{w_0}, \phi(\mathbf{x})\rangle})^2}\phi(\mathbf{x}) \otimes \phi(\mathbf{x})(\mathbf{w} - \mathbf{w}_0) + o_p(1)(\mathbf{w} - \mathbf{w}_0).$$

Using the same arguments as in the proof of Lemma 1, we can assert that $g(\mathbf{w}_n) = o_p(\frac{1}{\sqrt{n}})$. Now we can expand $\hat{L}(\mathbf{w}_n)$ in the neighborhood of $\mathbf{w}^*$ to get

$$\sqrt{n}P_n\nabla_{\mathbf{w}}l(\mathbf{x}, y; \mathbf{w}_n) = o_p(1)$$

$$=\sqrt{n}P_n\nabla_{\mathbf{w}}l(\mathbf{x}, y; \mathbf{w}^*) + \left(P_n\frac{e^{y\langle\mathbf{w}^*, \phi(\mathbf{x})\rangle}}{(1 + e^{y\langle\mathbf{w}^*, \phi(\mathbf{x})\rangle})^2}\phi(\mathbf{x}) \otimes \phi(\mathbf{x}) + o_p(1)\right)(\mathbf{w} - \mathbf{w}^*)$$

$$\stackrel{(i)}{=}\sqrt{n}P_n\nabla_{\mathbf{w}}l(\mathbf{x}, y; \mathbf{w}^*)$$

$$+ \sqrt{n}\left(P\frac{e^{y\langle\mathbf{w}^*, \phi(\mathbf{x})\rangle}}{(1 + e^{y\langle\mathbf{w}^*, \phi(\mathbf{x})\rangle})^2}\phi(\mathbf{x}) \otimes \phi(\mathbf{x}) + (P_n - P)\frac{e^{y\langle\mathbf{w}^*, \phi(\mathbf{x})\rangle}}{(1 + e^{y\langle\mathbf{w}^*, \phi(\mathbf{x})\rangle})^2}\phi(\mathbf{x}) \otimes \phi(\mathbf{x}) + o_p(1)\right)(\mathbf{w} - \mathbf{w}^*)$$

$$=\sqrt{n}P_n\nabla_{\mathbf{w}}l(\mathbf{x}, y; \mathbf{w}^*) + \sqrt{n}\left(P\frac{e^{y\langle\mathbf{w}^*, \phi(\mathbf{x})\rangle}}{(1 + e^{y\langle\mathbf{w}^*, \phi(\mathbf{x})\rangle})^2}\phi(\mathbf{x}) \otimes \phi(\mathbf{x}) + o_p(1)\right)(\mathbf{w} - \mathbf{w}^*),$$

where (i) is due to the law of large numbers. Because of $\forall\mathbf{x} : k(\mathbf{x}, \mathbf{x}) \leq 1$, the (weak) law of large numbers and the central limit theorem still holds for random variables in $\mathcal{H}$. For the former, we can prove using McDiarmid's inequality, symmetrization, and Jensen's inequality. For the latter, the proof is the same as the traditional proof of CLT using characteristic functions. Similarly, Slutsky's theorem still holds. As a result, the asymptotic distribution of $\sqrt{n}(\mathbf{w}_n - \mathbf{w}^*)$ is again Gaussian, with mean 0 and variance
$$\Sigma = \left(\mathbb{E}_{\mathbf{x}}[p_{\mathbf{w}^*}(1 - p_{\mathbf{w}^*})\phi(\mathbf{x}) \otimes \phi(\mathbf{x})]\right)^{-1},$$
where
$$p_{\mathbf{w}^*} = \frac{1}{1 + e^{-\langle\mathbf{w}^*, \phi(\mathbf{x})\rangle}}$$

$\square$

### A.3.1 ADDITIONAL EXAMPLES AND PROOFS FOR SECTION 4

**An example where** $p_{\text{learn}}(\mathbf{w}) = \mathcal{N}(\mathbf{w}^*, \sigma^2 I)$ Let $\text{Ber}_d(p)$ be the $d$ dimensional Bernoulli distribution where each dimension independently takes 1 with probability $p$ and $-1$ with probability $1 - p$.

Let $p^*(x, y)$ be defined by $p(y) = \text{Ber}_1(1/2)$ and $p(x|y = 1) = \text{Ber}_d(3/4)$ and $p(x| = 0) = \text{Ber}_d(1/4)$. We choose $w^* = \left(\frac{1}{2}, \frac{1}{2}, \cdots, \frac{1}{2}\right)$ and the ground truth score function is $g_{w^*}(x) = (x, w^*)$.

We draw a dataset of size $n$ from the distribution $\{x_1, y_1, \cdots, x_n, y_n\}$, to learn $\mathbf{w}$ the algorithm simply takes the average $\frac{\sum_{i:y_i=1}\mathbf{x}_i}{\sum_{i:y_i=1}1} - \frac{\sum_{i:y_i=-1}\mathbf{x}_i}{\sum_{i:y_i=-1}1}$. Then by CLT we have $p_{\text{learn}}(\mathbf{w}) = \mathcal{N}(\mathbf{w}^*, \sigma^2 I)$, where $\sigma^2 = 3/4n$. note that it is possible to obtain non-asymptotic results because $\mathbf{w}$ is actually distributed as a multinomial, but we will use the convenience of a Gaussian distribution.

**Proof of no-response rate**. When $\alpha = \sqrt{d}$ and $\tau = 0$ we have
$$\Pr_{p^*(x)}[\text{NR}(x)] \leq \Pr\left[|(x, w)| \leq \alpha\right] \leq \Pr\left[|(x, r^*)| \leq d/2 - \sqrt{d}\right]$$

$$= \Pr\left[|\mathcal{N}\left(0, d\sigma^2\right)| \leq d/2 - \sqrt{d}\right] \rightarrow_{d\to\infty} 0$$

Similarly for $\alpha = \sqrt{d}$ and $\tau = \frac{\alpha^2}{2\log 2/\delta}$ we have
$$\Pr_{p^*(x)}[\text{NR}(x)] \leq \Pr\left[|(x, w) + \mathcal{N}(0, \tau^2)| \leq \alpha\right] \leq \Pr\left[|(x, r^*) + \mathcal{N}(0, \tau^2)| \leq d/2 - \sqrt{d}\right]$$

$$= \Pr\left[\left|\mathcal{N}\left(0, d\sigma^2 + \frac{d}{2\log(2/\delta)}\right)\right| \leq d/2 - \sqrt{d}\right] \rightarrow_{d\to\infty} 0$$

