# OpenReview forum: "Towards Certified Defense for Unrestricted Adversarial Attacks"
_ICLR.cc/2020/Conference — Reject_

### Official Review · AnonReviewer1 · 2019-10-23
**Official Blind Review #1**

**Rating:** 3

**Review:**

The authors propose a new certified defense strategy that considers unrestricted black box attacks.  The paper provides bounds for the minimum number of queries needed for the attacker to attack the classifier successfully and then the authors prove that they can devise a defender to be robust against that attack. Here are a few points to consider:
1.	The paper is a bit difficult to understand (also not well-structured). The specific contributions are not quite clear with respect to the existing literature (which is reviewed in a sparse manner in the paper). Especially, the novelty of the theory and analysis presented here is a bit difficult to assess.
2.	 The results are not really validating the points they make in the analysis (for example in part 4.2 they talk about upper and lower bounds for number of queries as a function of τ, α, etc. but they never provide some plots or tables regarding that in the results section).
4.	Also, in Fig 2, it is hard to grasp the performance of the defended vs undefended classifiers with respect to the lower and upper bound that they have computed theoretically in the previous section.
5.	They emphasize on the “defensibility” and “query privacy” in the analysis but they do not provide anything in the results section considering them.
6.	As this is primarily a theoretical paper, focusing on simple classifiers is probably okay, but some sort of empirical comparison with other certified defense strategies is necessary. Just claiming that none of the existing methods would work for unrestricted attacks will not work is not sufficient. Some empirical results to show the specific advantages (e.g., at what budget the existing methods start to fail and the proposed method continues to perform well).

**Experience Assessment:**

I have published one or two papers in this area.

**Review Assessment: Checking Correctness Of Derivations And Theory:**

I assessed the sensibility of the derivations and theory.

**Review Assessment: Checking Correctness Of Experiments:**

I assessed the sensibility of the experiments.

**Review Assessment: Thoroughness In Paper Reading:**

I read the paper at least twice and used my best judgement in assessing the paper.

---

### Official Review · AnonReviewer3 · 2019-10-24
**Official Blind Review #3**

**Rating:** 1

**Review:**

Although this paper's title contains "certified defense" and "unrestricted adversarial attack",  what I believe this paper is doing is analyzing the query complexity of query-based black-box attacks under simple linear models such as logistic regressions (or kernelized versions). The authors considered a binary classifier with the additional capability of giving "no response" when the confidence is low.  In addition, the output of the classifier has to be perturbed by a random Gaussian vector. The authors then define several metrics including defensibility and query privacy to develop the query complexity on the considered model. The authors tested the query performance on two attacks: (1) the sign attack proposed by the authors and (2) the simba attack proposed by Guo et al.

I have several concerns regarding this paper:

1. In my perspective, the title is very misleading and does not properly justify the claims made in this paper. "Certified defense" usually refers to consistent top-1 prediction of a perturbed data sample under a defined threat model. The paper reads like the authors are actually certifying the defined defensibility metric but without a threat model to certify. In addition, the attack setting is limited to black-box attacks (i.e. zero-order adversary), whereas in certified defense the attack assumption is white-box.

2. It is also very unclear how unrestricted attack plays a role in the studied problem.  In the introduction, the authors' definition of adversarial examples is "any input is considered a valid adversarial example as long as it induces the classifier to predict a different label than an oracle classifier." But what is the oracle classifier? How do we justify the credibility of the "adversarial examples" in the experiments?

3. Only two black-box attacks were compared in this paper, one is the sign attack proposed by the authors, the other is the simba attack proposed by Guo et al. To my knowledge, simba attack paper has not been published at any peer-reviewed venue. In other words, both attacks are not widely recognized attacks or methods from published papers. Therefore, the performance evaluation is not fully justified. Since there are many black-box attacks from published papers, why not do performance analysis on those attacks?

4. Similar to 3, the classifier setting is also uncommon. Although I am happy to see classifiers have the ability to give no-response,  admittedly this type of classifier is rarely used in practice, not to mention the analysis is tied with Gaussian perturbation on the output. The technical contributions can be limited if the main contribution of this paper is characterizing the query complexity (or defensibility) of an uncommon classifier with Gaussian perturbation on the output. I believe providing more insights on how the analysis can be useful to mainstream classifiers are critical and necessary.

***Post-rebuttal comments
I thank the authors for the response. I hope the comments areuseful for preparing a future version of this work.
***

**Experience Assessment:**

I have published in this field for several years.

**Review Assessment: Checking Correctness Of Derivations And Theory:**

I carefully checked the derivations and theory.

**Review Assessment: Checking Correctness Of Experiments:**

I carefully checked the experiments.

**Review Assessment: Thoroughness In Paper Reading:**

I read the paper thoroughly.

---

### Official Review · AnonReviewer2 · 2019-10-27
**Official Blind Review #2**

**Rating:** 3

**Review:**

The paper proposes adding noise to the output of scoring function to defend from black-box attacks. This topic is actually very interesting so I enjoyed reading the paper, although it is currently only working for logistic regression and Naive Bayes and there are several unclear parts. I have several concerns about this paper, especially the claim of robust towards arbitrary perturbation.

- My main concern is about the assumption of the attacker. Based on the discussions in Section 3, it seems the authors assume that the query complexity of attacker relies on how many queries the attacker needs to recover a w that is close enough to w*. I don't think this is the correct assumption for the current attacks --- given an example, black box attacks are trying to find some x' for each x without trying to recover  or even estimate w. Therefore I wonder why the query complexity can be linked to the complexity of estimating w and is there any further assumption you need to make?

If the goal is to protect w, then this has been studied in several privacy/security papers and it's a different topic from adversarial attack. So the connection here is important but somehow unclear in the current draft.

- For the experiments, to justify it is robust to attack I think it's important to try on various black-box attacks, including ZOO (Chen et al., 2017), Natural evolution strategy (Ilyas et al., 2018), Nattack (Li et al, 2019). For decision-based black box settings Boundary attack (Brendel et al., 2018) and OPT-attack (Cheng et al., 2019).
(Not saying you should try all of them, but I feel more than 1 attack is needed to justify the claim).

- Some unclear points that need further clarification:

I feel assuming there's an optimal w* that correctly classifies data is unrealistic. Is is possible to relax this?

Condition 1: I fail to understand how is this related to q (attacker)? This seems only guaranteeing there's a majority mass of w centered at w*.

Condition 2: What is I ? (I didn't see the definition).

- Some related work:
In DNN defense there are some related work on adding random noise. In [1], I think they only require adding a random layer which can be in the final layer of network, corresponding to adding random to the scoring function. In [2], they assume adding randomness to each layer so only adding random to final layer is a special case of that. I know the guarantees here are very different from those papers, but it will be nice to have some discussions.

[1] "Certified Robustness to Adversarial Examples with Differential Privacy" Lecuyer et al., (S&P'19)
[2] "Towards Robust Neural Networks via Random Self-ensemble" Liu et al., (ECCV '18)

======

Thank you for the response and the additional experiments. I feel the paper has some interesting ideas and could be improved by a more careful writing and slightly adjusting the claim. I will rate the current draft borderline but slightly leaning to reject.

**Experience Assessment:**

I have published in this field for several years.

**Review Assessment: Checking Correctness Of Derivations And Theory:**

I assessed the sensibility of the derivations and theory.

**Review Assessment: Checking Correctness Of Experiments:**

I carefully checked the experiments.

**Review Assessment: Thoroughness In Paper Reading:**

I read the paper at least twice and used my best judgement in assessing the paper.

---

### Author Response · Authors · 2019-11-15
**Thank You for Your Suggestions and Comments**

We highly appreciate the reviewers for taking the time to provide helpful reviews and suggestions. We agree that the current writing can be organized better, and empirical results can be strengthened with additional experiments. Unfortunately, the rebuttal period is too short to address all these issues, so we would like to further improve the paper and submit to a future venue.

That being said, we firmly believe in the value of the framework proposed in our paper. In this response we would like to clarify several concerns about the framework.

Q: The threat model is non-standard

Response: Our approach is the standard black box attack setup. Attacker need to produce an input x that is mis-classified by the classifier. The difference is that we allow the classifier to answer “I don’t know”. This is necessary to defend against unrestricted attack. For example, the attacker could choose random noise as input; if a classifier outputs a prediction on invalid input—and the prediction is used for high risk decisions—this can be a security threat.

We show how to preserve privacy with respect to classifier weights w, but this is not the end goal.  Our main contribution is to associate knowledge about w with the attack’s ability to generate an adversarial example x (Theorem 1). Intuitively, if the attacker knows w, he or she can certainly generate an adversarial example; preventing any attacker from knowing w accurately is a way to prevent adversarial attack, and Theorem 1 precisely quantifies this.

Q: Meaning of “oracle classifier”

In our theoretical analysis, an oracle classifier is the optimal classifier that (globally) minimizes classification loss given infinite data.

Q: Defensibility (condition 1) does not mention attacker or threat model

We believe condition 1 should be unrelated to the attacker. Defensibility (condition 1) is solely a property of the classifier. This is necessary to provide guarantees on *any* attacker instead of a fixed attack.

Q: More experiments are needed.

Response: We performed additional experiments on NES (Ilyas et al, 2018) and Sign-OPT (Cheng et al, 2019). We observed similar results as Simba (Guo et al, 2019). We will include these results in the future submission. There is certainly a gap between theory (linear models) and deep neural networks; we will invest considerable effort in bridging this gap. We will include empirical analysis of the best theoretical guarantee, and comparison with other defense methods in the next revision.

---

### Decision · Program_Chairs · 2019-12-19

**Decision:**

Reject

**Comment:**

This paper proposes a certified defense under the more general threat model beyond additive perturbation. The proposed defense method is based on adding noise to the classifier's outputs to limit the attacker's knowledge about the parameters, which is similar to differential privacy mechanism. The authors proved the query complexity for any attacker to generate a successful adversarial attack. The main objection of this work is (1) the assumption of the attacker and the definition of the query complexity (to recover the optimal classifier rather than generating an adversarial example successfully) is uncommon, (2) the claim is misleading, and (3) the experimental evaluation is not sufficient (only two attacks are evaluated). The authors only provided a brief response to address the reviewers’ comments/questions without submitting a revision. Unfortunately none of the reviewer is in support of this paper even after author response.